



# Topographic uncertainty quantification for flow-like landslide models via stochastic simulations

Hu Zhao[1] and Julia Kowalski[1]

[1]RWTH Aachen University, Templergraben 55, 52062 Aachen, Germany

**Correspondence:** Hu Zhao (zhao@aices.rwth-aachen.de)

**Abstract.** Topography representing digital elevation models (DEMs) are essential inputs for computational models capable of simulating the run-out of flow-like landslides. Yet, DEMs are often subject to error, a fact that is mostly overlooked in landslide modeling. We address this research gap and investigate the impact of topographic uncertainty on landslide-run-out models. In particular, we will describe two different approaches to account for DEM uncertainty, namely unconditional and conditional

stochastic simulation methods. We investigate and discuss their feasibility, as well as whether DEM uncertainty represented by stochastic simulations critically affects landslide run-out simulations. Based upon a historic flow-like landslide event in Hong Kong, we present a series of computational scenarios to compare both methods using our modular Python-based workflow. Our results show that DEM uncertainty can significantly affect simulation-based landslide run-out analyses, depending on how well the underlying flow path is captured by the DEM, as well as further topographic characteristics and the DEM error's variability.

We further find that in the absence of systematic bias in the DEM, a performant root mean square error based unconditional stochastic simulation yields similar results than a computationally intensive conditional stochastic simulation that takes actual DEM error values at reference locations into account. In all other cases the unconditional stochastic simulation overestimates the variability of the DEM error, which leads to an increase of the potential hazard area as well as extreme values of dynamic flow properties.

**Keywords**: flow-like landslide, run-out modeling, topographic uncertainty, stochastic simulation, hazard analysis

## 1   Introduction

Landslides are natural hazards that occur frequently all around the world causing casualties, economic devastation, and environmental destruction. Most often, they are naturally driven, e.g. by means of long-lasting and/or intensive precipitation events, or induced by earthquakes. Yet, landslides might also be triggered or its susceptibility increased as a result of human

activities, e.g. deforestation and construction. According to the United Nations Office for Disaster Risk Reduction and the Center for Research on the Epidemiology of Disasters, 378 recorded landslides from 1998 to 2017 affected 4.8 million people and caused 18414 deaths as well as several billion US dollars of economic losses (Wallemacq et al., 2018). Froude and Petley (2018) reported that in total 55997 people were killed during 4862 fatal non-seismic landslide events from January 2004 to December 2016. Still, it has to be assumed that the damage potential of landslides is underestimated as 1) events have been





under-reported for decades, especially in developing countries, and 2) losses caused by co-seismic landslide events tend to be
classified as secondary losses due to earthquakes.

Rapid flow-like landslides, such as rock avalanches and debris flows, show a particularly high hazard potential due to their
high mobility, long travel distance and fast propagation speed. In recent years, the geo-hazard community put a lot of effort
into developing computational run-out models in order to assess and predict risks associated with rapid landslides and to
develop mitigation strategies. Most of the models in practical use are based on a (computationally efficient) 'shallow flow
type' process description and depth-averaging techniques (e.g. Pitman et al., 2003; Hungr, 2009; Pastor et al., 2009; Christen
et al., 2010; Xia and Liang, 2018). In these, the flowing material is treated as an 'equivalent fluid' and governed by idealized
internal and basal rheologies (Hungr, 2009). Alternative (computationally demanding) models aim at a direct description of
fully three-dimensional flow behavior. They hence offer a higher process complexity level (e.g. Mast et al., 2014; Teufelsbauer
et al., 2011), yet are typically not feasible for practical hazard mitigation purposes. Detailed reviews of computational run-out
models for rapid, flow-like landslide models have been published by McDougall (2017) and Pastor et al. (2018).

An indispensable input to any of these computational landslide run-out models is data that represents the terrain in which
the slide is likely to occur. Pioneered by Miller and Laflamme (1958), digital elevation models (DEMs) have become the
most popular form of representing topographies in the scientific community. Methods for generating DEMs have evolved
rapidly over decades from conventional approaches like field surveying and topographic map digitizing, to passive and active
remote sensing, such as stereoscopic photogrammetry, interferometric synthetic aperture radar (InSAR), and light detection and
ranging (LiDAR), see Wilson (2012) for a comprehensive review. Differences between these methods exist in their footprint,
cost, resolution and accuracy of the resulting DEM. Whatever method used, however, the resulting DEM will inevitably contain
errors that are introduced either during source data acquisition or during data processing. The so-called DEM error hence refers
to the difference between the true real world elevations and their DEM representation. Typically, there is a lack of information
on the DEM error, which led to notion of 'DEM uncertainty' that refers to what we do not know about the error, see Wechsler
(2007).

Nowadays, several global DEM databases, e.g. SRTM (Rodriguez et al., 2006), AW3D30 (Courty et al., 2019), and TanDEM-
X (Wessel et al., 2018), as well as some regional DEM databases (Pakoksung and Takagi, 2016) are publicly available. Also
commercial DEM databases exist that are associated with significant costs (Hawker et al., 2018). Online initiatives such as
*OpenTopography* facilitate community access and aim at democratizing online availability of high-resolution topography data
acquired with LiDAR and other technologies (Krishnan et al., 2011). Despite the broad variety of existing DEM sources,
however, we are still facing (and will face in the near future) a very limited availability of high-accuracy DEMs for some
regions that are particularly prone to landslide hazards, e.g. in Asia (Froude and Petley, 2018). Whenever using DEM data for
simulation-based landslide hazard analysis, it is hence important to be aware of DEM error and uncertainty, and to consider its
potential impact on computational run-out analyses and related computational risk assessments.

DEM error has arisen researchers' attention since long. Many efforts have for instance been put into quantifying the error
associated with specific DEM sources based on data of higher accuracy, e.g. acquired by satellite measurements (Berry et al.,
2007; Mouratidis and Ampatzidis, 2019), medium footprint LiDAR (Hofton et al., 2006), or GPS survey (Rodriguez et al.,





2006; Bolkas et al., 2016; Patel et al., 2016; Wessel et al., 2018; Elkhrachy, 2018). Meanwhile, a variety of methods have been devised to classify DEM error into various categories (Oksanen, 2003; Hengl et al., 2004; Fisher and Tate, 2006). Due to the complexity of potential influencing factors (sensor technology, retrieval algorithms, data processing, land cover and surface morphology, terrain attributes (Wilson, 2012; Fisher and Tate, 2006; Gonga-Saholiariliva et al., 2011)), these methods can only constrain the DEM error, and will not deterministically correct for it at all grid points. Hence, DEM uncertainty remains, and

has to be accounted for in any subsequent analysis that relies on the DEM data.

In this circumstance, a stochastic simulation is an effective computational approach to deal with the situation (Holmes et al., 2000). Instead of considering a single (assumed as accurate) DEM, the fundamental idea of a stochastic simulation in the context of DEM uncertainty propagation is to separate the DEM into a known deterministic contribution and an unknown DEM error. DEM uncertainty is then accounted for by treating the DEM error as a random field consisting of a collection of random

variables defined at selected grid points. An ensemble of equiprobable realizations of the random field is then generated based on certain assumptions and available information of DEM error. This could for instance be the so-called root mean square error (RMSE), a minimalistic indicator for the overall error magnitude or a semivariogram that informs about the spatial autocorrelation of the DEM error. Adding the DEM error realizations to the known deterministic DEM contribution results in an ensemble of equiprobable DEM realizations, which can finally be used for a DEM uncertainty propagation analysis.

Stochastic simulation methods for DEM uncertainty propagation analyses have been developed since the 1990s and are by now widely applied in many fields, including terrain analysis (Holmes et al., 2000; Raaflaub and Collins, 2006; Moawad and EI Aziz, 2018), flood modeling (Watson et al., 2015; Hawker et al., 2018; Kiczko and Miroslaw-Swiatek, 2018), soil erosion modeling (Aziz et al., 2012), landslide susceptibility mapping (Qin et al., 2013), dry block and ash flow modeling (Stefanescu et al., 2012), etc. With respect to rapid, flow-like landslide run-out modeling, very little work has been done to assess the

potential impact of DEM uncertainty, most likely due to the complexity, and hence level of sophistication of the associated process models. Meanwhile, however, advances in computing technology led to computationally feasible and well-developed landslide run-out simulation tools. As one of the most important inputs for these tools, a DEM determines the landslide's flow path. A natural next step is hence to consider the impact of DEM uncertainty in these models, as overlooking DEM uncertainty may lead to a bias of risk management decisions in a wrong direction. The major aim of this study is therefore to describe two

different approaches in order to incorporate DEM uncertainty into computational landslide run-out analyses, and to investigate and discuss their feasibility, as well as whether DEM uncertainty is critical to landslide run-out and affects its results.

This paper is organized as follows: In section 2, we briefly describe the landslide run-out model used in this study, which is a continuum-mechanical shallow flow model based on the Voellmy-Salm rheology. In section 3, we recall on various methods to account for DEM uncertainty with a major focus on two approaches, namely an unconditional and a conditional stochastic

simulation method. The rest of the paper is devoted to investigating DEM uncertainty propagation for rapid, flow-like landslides based on an integrated workflow that combines the aforementioned computational process model (section 2) with the stochastic DEM simulations (section 3). Note, that while in our particular study we chose to use a continuum-mechanical shallow flow process model based on the Voellmy-Salm rheology, the workflow itself is modular and non-intrusive. It would hence also possible to couple the stochastic DEM simulation with any other (DEM based) computational landslide model. Section 4



describes the modular Python-based workflow that we developed in order to set-up and manage the workflow and to interpret its simulation results. We present a series of computational scenarios based upon a historic landslide event in section 5. All scenarios compare the unconditional and conditional stochastic DEM simulation. Finally, section 6 is devoted to a discussion of our results. Important conclusions are drawn in section 7.

## 2 Landslide process model

As detailed in the introduction, a variety of process-based computational landslide run-out models have been developed in recent decades. Among these is a family of depth-integrated shallow flow type landslide models that we chose as the basis for our work. Shallow flow type landslide models can be further classified based on their applied basal rheology, e.g. Voellmy, Bingham, Quadratic resistance model, etc. (Naef et al., 2006; Hungr and McDougall, 2009). Our study uses the Voellmy-Salm (VS) process model, which is a depth-averaged continuum mechanical model incorporating the Voellmy basal rheology. Note that the stochastic workflow presented later is modular and does not depend on this choice. Hence, the Voellmy model can straight-forwardly be substituted by another computational process models.

### 2.1 Reference frame and relation to topographic error

Let $\{X, Y, Z\}$ denote a fixed Cartesian coordinate system, in which $X$ and $Y$ are the horizontal axes and $Z$ is the vertical axis. The coordinates of a point in the Cartesian coordinate system are denoted by $(X, Y, Z)$. A topography can then be expressed as a surface mapping of horizontal $X$ and $Y$ coordinates and represents the elevation at each point, namely $Z(X, Y)$. The mapped topography induces a surface coordinate system $\{x, y, z\}$, in which $x$ and $y$ denote tangential directions and $z$ points in the direction of the surface normal. Any vector that is constant with respect to the fixed Cartesian coordinates system, e.g. gravitational acceleration $\boldsymbol{g} = (g_X, g_Y, g_Z)^T = (0, 0, -g)^T$, hence spatially varies, when interpreted in terms of the surface mapped coordinated system $\boldsymbol{g} = (g_x, g_y, g_z)^T$. Error or uncertainty in the topography representation $Z(X, Y)$ hence directly translates into error and uncertainty of that vector representation.

### 2.2 Voellmy rheology computational process model

The Voellmy process model along with its computational implementation is described in Bartelt et al. (1999) and Christen et al. (2010). It assesses the slide's dynamics in terms of flow height $H(x, y, t)$ and depth-averaged velocity $\boldsymbol{U}(x, y, t) := (U_x(x, y, t), U_y(x, y, t))^T$, both of which depend on time $t$ and spatial coordinates $x$ and $y$. The governing system reads

$$\partial_t H + \partial_x(HU_x) + \partial_y(HU_y) = \dot{Q}(x, y, t)$$

$$\partial_t(HU_x) + \partial_x\left(HU_x^2 + g_z\frac{H^2}{2}\right) + \partial_y(HU_xU_y) = g_xH - n_x(\mu g_z H + g\|\boldsymbol{U}\|^2/\xi)$$

$$\partial_t(HU_y) + \partial_x(HU_xU_y) + \partial_y\left(HU_y^2 + g_z\frac{H^2}{2}\right) = g_yH - n_y(\mu g_z H + g\|\boldsymbol{U}\|^2/\xi)$$

(1)





Here, the first equation denotes the mass balance, in which $H$, $U_x$ and $U_y$ stand for height and surface tangential velocity components, and $\dot{Q}(x,y,t)$ stands for a mass production source term that accounts for erosion of material along the way. Second and third equations denote the $x$ and $y$ momentum balance, in which $g_x$, $g_y$, and $g_z$ are the three local components of gravitational acceleration vector $\boldsymbol{g}$. Furthermore, $n_x$ and $n_y$ are $x$ and $y$ components of the unit vector $\boldsymbol{n}$ that opposes

the local velocity, and $\mu$ and $\xi$ are two friction parameters that stand for dry Coulomb and 'turbulent' friction coefficients respectively. The friction parameters are determined by back-analysis based on historic events. Note that additional model parameters introduced in the original publications, such as velocity shape factors and non-hydrostatic pressure corrections are not taken into account as they are hard to constrain and have been shown to not critically affect the slide's dynamics (e.g. Hungr et al., 2005; Christen et al., 2010).

The topographic surface $Z(X,Y)$ enters the governing equations of the process model implicitly in terms of the spatially varying gravitational acceleration vector $\boldsymbol{g} = (g_x, g_y, g_z)^T$. Any error and uncertainty present in the topography representation hence also enters the landslide run-out simulation results.

The VS model had been first proposed to model snow avalanche (Salm, 1993). Nowadays, it has been widely applied to other types of gravity-driven rapid mass movements including flow-like landslides (Pastor et al., 2018; Frank et al., 2015; Hussin

et al., 2012; Kumar et al., 2019). In this study, the proprietary mass flow simulation platform RAMMS (Christen et al., 2010) which provides a GIS integrated implementation of the VS model is used for landslide run-out modeling. It is integrated as a module of our workflow (see section 4) that is developed for the purpose of DEM uncertainty propagation analysis.

## 3 Methods to represent DEM uncertainty

Again, the topographic surface is expressed as a function $Z(X,Y)$ parametrized in horizontal coordinates $X$ and $Y$. In practice,

the function $Z(X,Y)$ is often constructed from discrete gridded raster data. We hence assume that a domain of interest $D$ is discretized into the horizontal $X$ and $Y$ direction, which results in a spatial grid defined as

$$\boldsymbol{D}_{mn} = \{D_{ij} = (X_i, Y_j) \mid (X_i, Y_j) \in D; \ i = 1, 2, ..., m; \ j = 1, 2, ..., n\}. \tag{2}$$

The elevation data associated with each grid point $D_{ij}$ is defined as

$$\boldsymbol{Z}_{mn} = \{Z_{ij} = Z(X_i, Y_j) \mid \forall \, D_{ij} \in D_{mn}\}. \tag{3}$$

The elevation $\boldsymbol{Z}_{mn}$ of a common DEM data product might be erroneous with respect to the true values as discussed in the introduction. If we denote the true elevation as

$$\boldsymbol{Z}^*_{mn} = \{Z^*_{ij} = Z^*(X_i, Y_j) \mid \forall \, D_{ij} \in D_{mn}\}, \tag{4}$$

the DEM error can be expressed as

$$\boldsymbol{\epsilon}_{mn} = \{\epsilon_{ij} = Z^*_{ij} - Z_{ij} \mid \forall \, D_{ij} \in D_{mn}\}. \tag{5}$$





If we knew the error $\epsilon_{mn}$, we would be able to recover the real world topographic surface $\boldsymbol{Z}^*_{mn}$. The fact, however, that the error

is unknown, or we only have limited information about the error implies an uncertainty to the input of our landslide process

simulation. Within this study, we will refer to the uncertainty associated to the unknown DEM error as DEM uncertainty. In

this circumstance, each $\epsilon_{ij}$ is treated as a random variable and $\epsilon_{mn}$ is accordingly treated as a random field, which consists

of a collection of random variables $\epsilon_{ij}$. By generating multiple realizations of the random field $\epsilon_{mn}$, DEM uncertainty can

be represented. This process is widely known as stochastic simulation. It requires a suitable model to describe the jointed

uncertainty of all $\epsilon_{ij}$ based on limited available information of DEM error. The task can be further divided into determining:

1) the probability distribution function (pdf) of each $\epsilon_{ij}$ which quantifies local uncertainty at each grid point; 2) the correlation

between different $\epsilon_{ij}$ which is usually termed as spatial autocorrelation of DEM error.

According to available information on the DEM error, existing approaches that could be used to solve the two issues can be

generally classified into two groups:

A) unconditional stochastic simulation (USS);

B) conditional stochastic simulation (CSS).

More specifically, USS is only informed with properties of DEM error, e.g. the RMSE, and thus does not honour any actual

DEM error values. CSS is informed with certain number of actual DEM error values at reference locations within the DEM,

e.g. obtained from higher accurate reference data, and thus could directly honour the actual DEM error values at reference

locations (Fisher and Tate, 2006).

### 3.1    Unconditional stochastic simulation (USS) based on the RMSE

Typically available information about the DEM error provided by DEM vendors is the root mean square error (RMSE). For a

set of $K$ reference locations, it is defined as

$$\text{RMSE} = \sqrt{\frac{1}{K}\sum_{k=1}^{K}(Z^*_{kk} - Z_{kk})^2}. \tag{6}$$

Here, $\boldsymbol{Z}^*_{KK} = \{Z^*_{kk} = Z^*(X_k, Y_k) \mid (X_k, Y_k) \in D; \; k = 1, 2, ..., K\}$ denotes higher accurate elevation values measured at the

reference locations and $\boldsymbol{Z}_{KK} = \{Z_{kk} = Z(X_k, Y_k) \mid (X_k, Y_k) \in D; \; k = 1, 2, ..., K\}$ denotes corresponding elevation values

based on the DEM.

It should be noted that while the RMSE is typically available, this is not true for the reference elevation values $\boldsymbol{Z}^*_{KK}$ itself. As

stated numerous times in the literature, it is critical that the RMSE only provides a global indication of DEM error magnitude

without any information about its spatial autocorrelation. Still, it is by far the most widely used DEM error indicator for many

DEM databases and mostly the only available information coming along with DEM products. In this circumstance, USS could

be used to represent DEM uncertainty and study its propagation into landslide run-out analyses.

Modeling DEM uncertainty based on USS assumes that all local error values $\epsilon_{ij}$ are independent and fulfill the same uni-

variate Gaussian distribution with a mean ($\mu$) of zero and a standard deviation ($\sigma$) equivalent to the given RMSE. Under this





assumption, an ensemble of spatially uncorrelated realizations of the random field $\boldsymbol{\epsilon}_{mn}$ can be generated by randomly assigning error values to each $\epsilon_{ij}$ according to its local Gaussian probability distribution.

In the next step, we have to account for the (unknown) spatial autocorrelation of $\boldsymbol{\epsilon}_{mn}$. Potential methods that could be applied are simulated annealing, spatial autoregressive modeling, spatial moving averages, etc., see Wechsler (2007). Simulated

annealing is generally computationally intensive and spatial autoregressive modeling becomes impractical for simulation of large areas (Oksanen, 2006). In this study, we use the spatial moving averages method that increases the spatial autocorrelation by filtering spatially uncorrelated realizations with a distance-weighted filter proposed by Wechsler and Kroll (2006). For $\epsilon_{ij}$ at one grid point of an uncorrelated realization, its value is replaced by the weighted average of $\epsilon_{ij}$ at all grid points within the filter kernel. The weight decreases with increasing of the distance to the grid point, which is similar to semivariogram trends

(Wechsler and Kroll, 2006). The size of the filter denoted as d depends on the maximum autocorrelation length of $\boldsymbol{\epsilon}_{mn}$ which again is unknown if the RMSE is the only available information. In practice, d is often determined based on the maximum autocorrelation length of the original DEM (Wechsler, 2007; Aziz et al., 2012).

Though it relies on some assumptions, such as an appropriate choice of correlation length d, the sketched approach is generally applicable if RMSE is the only available information. It may become critical if a DEM contains a systematic bias

which means that the mean of $\boldsymbol{Z}_{kk}^{*} - \boldsymbol{Z}_{kk}$ deviates largely from zero. More specifically, if we follow Fisher and Tate (2006) and Wessel et al. (2018) in defining mean $\mu$ and standard deviation $\sigma$ as

$$\mu = \frac{1}{K} \sum_{k=1}^{K} (Z_{kk}^{*} - Z_{kk}) \quad \text{and} \quad \sigma = \sqrt{\frac{1}{K-1} \sum_{k=1}^{K} ((Z_{kk}^{*} - Z_{kk}) - \mu)^2}, \tag{7}$$

we can express the RMSE in terms of $\mu$ and $\sigma$ as

$$\text{RMSE} = \sqrt{\mu^2 + \frac{K-1}{K} \sigma^2}. \tag{8}$$

If the number of reference points $K$ is relatively large, $\sqrt{(K-1)/K}$ is close to one. Equation (8) then indicates that the RMSE is larger than the standard deviation $\sigma$ if the mean $\mu$ deviates from zero. The difference between the RMSE and $\sigma$ increases with increasing $\mu$. For example, the $\mu$, $\sigma$, and RMSE of the global TanDEM-X DEM based on about three million reference points are $0.17\,m$, $1.28\,m$, and $1.29\,m$ (Wessel et al., 2018). That of the EU-DEM of Central Macedonia based on 12943 reference points are $1.8\,m$, $3.6\,m$, and $4.0\,m$ while that of the ASTER GDEM of the same area based on the same reference points are

$6.8\,m$, $7.6\,m$, and $10.2\,m$ (Mouratidis and Ampatzidis, 2019). This means that assuming the standard deviation of the DEM error to be given as the RMSE consequently overestimates the variability of the DEM error if the mean deviates largely from zero.

The implications of both issues, namely the fact that the filter size d is unknown and has to be subjectively chosen, and that the RMSE provides an insufficient representation of the DEM error, are investigated in the following study. For convenience,

the two issues are referred to as:

- *unrepresentative RMSE*,

- *subjective d*.





## 3.2 Conditional stochastic simulation (CSS) based on higher accurate reference data

This approach requires the availability of higher accurate reference data at certain reference locations, e.g. from higher accurate
DEM products, or GPS surveys. Note, that although these data might be subject to error themselves, it is fair to assume this
error to be much smaller. This justifies to use the higher accurate reference data as true elevation values $\boldsymbol{Z}^*_{KK}$. Based on $\boldsymbol{Z}^*_{KK}$,
we could determine the statistics of the DEM error, e.g. the RMSE, the $\mu$ and the $\sigma$ as discussed in section 3.1. Likewise, we
can assess the spatial autocorrelation of the DEM error, e.g. in the form of a semivariogram model (see Fig. 4). In addition, we
know the DEM error at the reference locations exactly, denoted as $\boldsymbol{\epsilon}^*_{KK} = \{ \epsilon^*_{kk} \mid k = 1, 2, ..., K \}$. Yet, we still lack knowledge
about the DEM error away from the $K$ reference locations, hence the complete random field $\boldsymbol{\epsilon}_{mn}$.

In that situation, conditional stochastic simulation (CSS) can be used to simulate, i.e. generate realizations of the random
field $\boldsymbol{\epsilon}_{mn}$. Many geostatistical methods of conditional simulation could be applied, including sequential simulation algorithms,
p-field approach, simulated annealing, etc. (Goovaerts, 1997). In this study, we apply a sequential Gaussian simulation. It is
the most attractive technique for stochastic spatial simulation according to Temme et al. (2009) and has been widely utilized in
DEM uncertainty propagation analysis (Holmes et al., 2000; Aziz et al., 2012).

The sequential Gaussian simulation sequentially samples each local error $\epsilon_{ij}$ along a random path that consists of all grid
points $D_{ij}$ under the multi-Gaussian assumption. This means that assuming the random field $\boldsymbol{\epsilon}_{mn}$ to satisfy a multivariate
Gaussian distribution, hence each $\epsilon_{ij}$ fulfills a univariate Gaussian distribution denoted as $N(\mu_{ij}, \sigma_{ij})$. The essential idea
now is that the mean $\mu_{ij}$ and standard deviation $\sigma_{ij}$ are determined sequentially by means of simple kriging based on: the
semivariogram model of DEM error that provides covariances in simple kriging equations, and the conditioning information
including $\boldsymbol{\epsilon}^*_{KK}$ and previously sampled $\epsilon_{ij}$. By making each univariate Gaussian distribution conditional not only to $\boldsymbol{\epsilon}^*_{KK}$ but
also to all previously sampled $\epsilon_{ij}$, the semivariogram model of DEM error is reproduced in realizations of $\boldsymbol{\epsilon}_{mn}$ (Goovaerts,
1997). The process to generate one realization of $\boldsymbol{\epsilon}_{mn}$ is as follows:

1) determine a semivariogram model to represent the spatial autocorrelation of DEM error based on normal score trans-
formed $\boldsymbol{\epsilon}^*_{KK}$;

2) define a random path visiting each $D_{ij}$ once;

3) at each $D_{ij}$, determine $N(\mu_{ij}, \sigma_{ij})$ using simple kriging based on the semivariogram model and normal score trans-
formed $\boldsymbol{\epsilon}^*_{KK}$;

4) sample a value from $N(\mu_{ij}, \sigma_{ij})$, assign it to $\epsilon_{ij}$, and add $\epsilon_{ij}$ into normal score transformed $\boldsymbol{\epsilon}^*_{KK}$;

5) repeat steps 3) and 4) until all $D_{ij}$ along the path are visited;

6) back-transform all sampled $\epsilon_{ij}$ to the original distribution of $\boldsymbol{\epsilon}^*_{KK}$.

Multiple realizations can be generated by defining different random paths.



## 4   Implementation

Studying the impact of DEM uncertainty on landslide run-out modeling is computationally intensive and technically demand-
ing. It includes representing DEM uncertainty in terms of a large number of DEM realizations, conducting numerous landslide
run-out modeling based on the DEM realizations, and postprocessing extensive output data. In addition, understanding how
DEM uncertainty affects terrain attributes may facilitate us to interpret its impact on landslide run-out modeling. This re-
quires the ability to calculate terrain attributes, e.g. slope, ruggedness, etc. of the original DEM as well as the generated DEM
realizations.

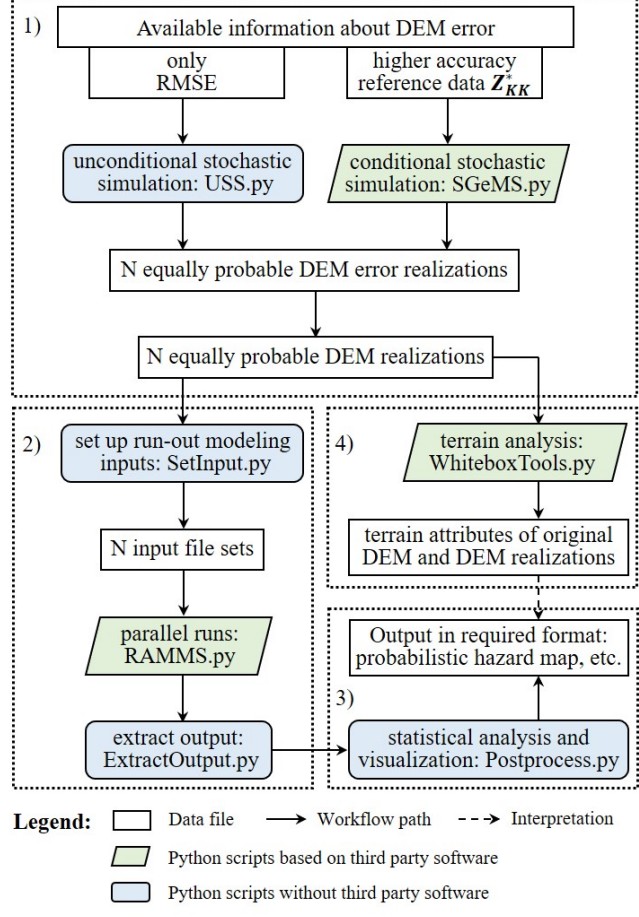

**Figure 1.** Computational workflow of DEM uncertainty propagation in landslide run-out simulation. It is part of our PSI-slide package
in development that is designed for the purpose of systematically investigating the impact of various sources of uncertainty on simulat-
ing gravity-driven mass movements (Kowalski et al., 2018; Zhao and Kowalski, 2018). The workflow consists of four modules: 1) DEM
uncertainty representation; 2) landslide run-out modeling; 3) statistical analysis and visualization; 4) terrain analysis.





In this study, we propose a workflow that integrates our own Python implementation of selected aspects of the workflow and existing software as well as toolboxes to solve above mentioned tasks. It is part of our PSI-slide (Predictive Simulation of Slides) package in development that is designed for the purpose of systematically investigating the impact of the various sources of uncertainty on simulating gravity-driven mass movements (Kowalski et al., 2018; Zhao and Kowalski, 2018). Herein, we focus on DEM uncertainty. Figure 1 illustrates the workflow. It consists of four modules:

1) DEM uncertainty representation. In this module, we generate an ensemble of N equally probable DEM realizations to represent DEM uncertainty based on available information about DEM error. USS as introduced in section 3.1 is implemented without third party software (USS.py) for cases in which only the RMSE is available. For cases in which higher accurate reference data is provided, CSS as introduced in section 3.2 is implemented by integrating the sequential Gaussian simulation algorithm of the Stanford Geostatistical Modeling Software (SGeMS) (Remy et al., 2009) into our
260         workflow (SGeMS.py).

    2) Landslide run-out modeling. This module is used to conduct N landslide run-out simulations based on the N DEM realizations generated in module 1). In this study we employ the proprietary mass flow simulation platform RAMMS (Christen et al., 2010) which provides a GIS integrated implementation of the VS model. First, a Python script named SetInput.py is called to set up required inputs for each simulation run. Then a Python script named RAMMS.py starts
265         parallel runs of RAMMS using the Python Scoop module. In the end, a Python script named ExtractOutput.py is called to extract user-specified outputs.

    3) Statistical analysis and visualization. This module is used to conduct statistical analysis on the user-specified outputs from module 2) and visualize results. It is mainly based on the Python Numpy and Matplotlib modules. For example, probabilistic hazard map can be produced to indicate potential hazard area.

4) Terrain analysis. This module is used to analyze terrain characteristics of the original DEM and DEM realizations from module 1), which may help us to interpret outputs from module 3). This is achieved by integrating several terrain analysis tools from WhiteboxTools (Lindsay, 2018) like calculating slope, aspect, ruggedness index, etc. into our workflow (WhiteboxTools.py).

## 5   Case study

This study is based upon a historic landslide and two DEM sources. For the purpose of DEM uncertainty propagation analysis, we assume one DEM source to be more accurate than the other and then obtain higher accurate reference data from the more accurate DEM source to assess elevation error of the less accurate DEM source. We design a series of computational scenarios based on the higher accurate reference data to study the impact of DEM uncertainty on landslide process simulation for both the case when only the RMSE is available and the case when higher accurate reference data is available. Additional computational
scenarios are designed to study the *unrepresentative RMSE* and *subjective d* issues as detailed in section 3.1 in the form of a sensitivity analysis.




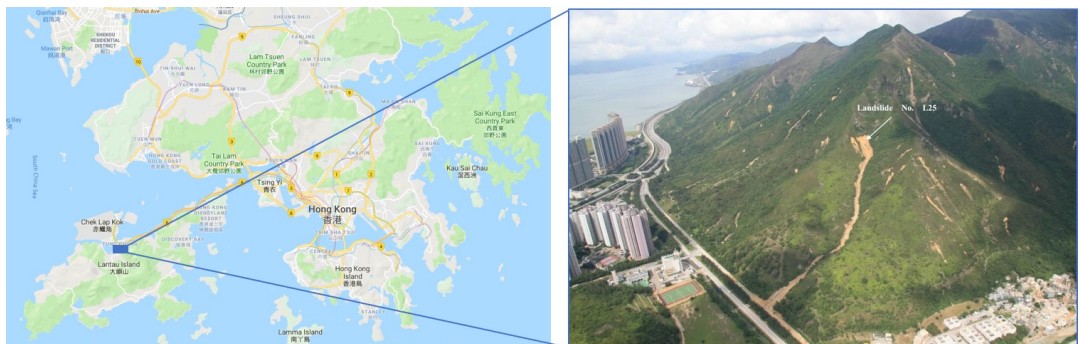

**Figure 2.** The 2008 Yu Tung Road landslide. Left: © Google Maps 2019 of Hong Kong (map data©2019); right: aerial photograph of Yu Tung Road site after the 2008 landslide. It corresponds to the No. L25 landslide in the GEO report (AECOM Asia Company Limited, 2012).

## 5.1 Scenario background and DEM sources

The historic landslide happened on June 7 2008 on the hillside above Yu Tung Road in Hong Kong due to an intense rainfall event, see Fig. 2. It was the largest flow-like landslide out of 19 landslides during that event. Around 3400 $m^3$ material were
mobilized and traveled about 600 $m$ until deposit. The landslide event had a severe infrastructural impact, as it led to closure of westbound lanes of Yu Tung Road for more than two months (AECOM Asia Company Limited, 2012). The Yu Tung Road landslide also served as a benchmark case for predictive landslide run-out analysis at the second Joint Technical Committee on Natural Slopes and Landslides (JTC1) Workshop on Triggering and Propagation of Rapid Flow-like Landslides in Hong Kong 2018 (Pastor et al., 2018). Two types of DEM data of the Yu Tung Road area had been the basis for this study:

- A public 5 $m$ resolution digital terrain model covering the whole area of Hong Kong (HK-DTM). It is downloaded from the website of the Survey and Mapping Office of Hong Kong. The HK-DTM is generated from a series of digital orthophotos, which are derived from aerial photographs taken in 2014 and 2015. The reported accuracy is ±5 $m$ at 90% confidence level. (DATA.GOV.HK, 2019)

- A 2 $m$ resolution DEM (2m-DEM) covering the main area of the Yu Tung Road landslide event. Its boundary is shown
in Fig. 3 (a). It had been provided for the benchmark exercise during the second JTC1 workshop. It is produced based on field mapping after the 2008 Yu Tung Road landslide event.

In this study, we assume the 2m-DEM to be more accurate than the 5 $m$ resolution HK-DTM. Similar to our consideration at the beginning of section 3.2, the 2m-DEM and 5 $m$ resolution HK-DTM correspond to $Z^*$ and $Z$ as defined in section 3. A set of higher accurate reference data $Z^*_{KK}$ can hence be determined to provide information to represent uncertainty of the 5 $m$
resolution HK-DTM.

It should be noted that the 2m-DEM and 5 $m$ resolution HK-DTM were produced in different time periods. After the 2008 Yu Tung Road landslide, debris-resisting barriers and a road had been built in the area within the red circle and blue rectangle in Fig. 3 (a) respectively. They are reflected in the HK-DTM but not in the 2m-DEM, which leads to large inconsistency between





the two DEMs in that area. Therefore, to avoid unrealistically large error of the HK-DTM, data from the 2m-DEM in that area

is excluded from higher accurate reference data $\boldsymbol{Z}_{KK}^*$.

## 5.2  DEM realizations

### 5.2.1  Information of DEM error

As shown in Fig. 3 (a), we evenly pick 180 reference locations from the HK-DTM grid points within the boundary of the 2m-DEM. Higher accurate reference data at these locations is obtained from the 2-m DEM using bilinear interpolation, denoted as

$\boldsymbol{Z}_{KK}^*\{K=180\}$. Subtracting the corresponding elevation values of the HK-DTM $\boldsymbol{Z}_{KK}\{K=180\}$ from $\boldsymbol{Z}_{KK}^*\{K=180\}$, we obtain elevation error values of the HK-DTM at the 180 reference locations, denoted as $\boldsymbol{\epsilon}_{KK}^*\{K=180\}$.

   Figure 3 (b) shows the histogram of $\boldsymbol{\epsilon}_{KK}^*\{K=180\}$. 90% of the elevation error values are within -5.84 m and $-1.04\,m$, which is close to the reported accuracy (see section 5.1). The $\mu$, $\sigma$, and RMSE according to Eq. (7) and Eq. (6) are -3.0 m, 1.5 m, and 3.3 m respectively. Here, it should be noted that the RMSE is larger than the $\sigma$ since the $\mu$ is not zero which indicates

a systematic bias. As discussed in section 3.1, this also indicates that assuming the standard deviation of the HK-DTM error being equivalent to the RMSE in USS would overestimate the variability of the HK-DTM error.

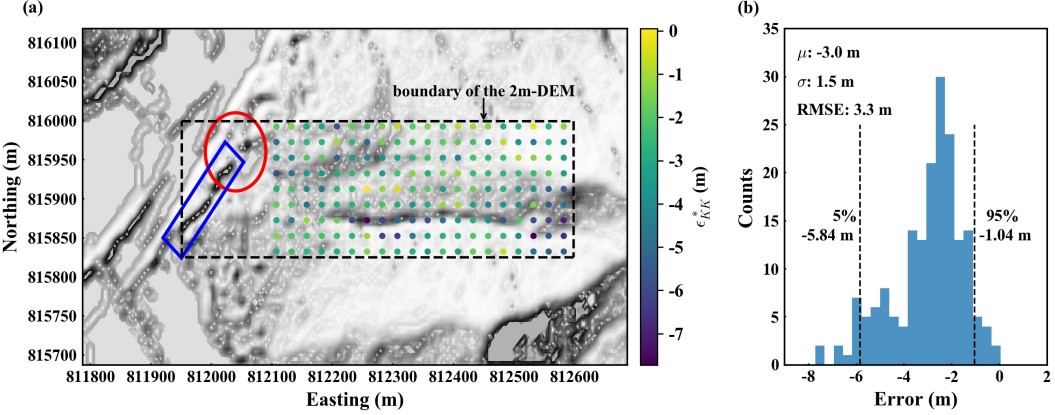

**Figure 3.** (a) Elevation error $\boldsymbol{\epsilon}_{KK}^*\{K=180\}$ of the HK-DTM at 180 reference locations. The background is the hillshade plot of the HK-DTM. Debris-resisting barriers and a road in the circle and rectangle area constructed after the 2008 landslide event are represented in the HK-DTM but not in the 2m-DEM. It causes inconsistency between the two DEMs in that area. To avoid unrealistically large error of the HK-DTM, data from the 2m-DEM in that area is excluded from higher accurate reference data. (b) Histogram of $\boldsymbol{\epsilon}_{KK}^*\{K=180\}$. The RMSE is larger than the standard deviation ($\sigma$) since the mean ($\mu$) is not zero. As discussed in section 3.1, this indicates that assuming the standard deviation of the HK-DTM error being equivalent to the RMSE in USS would overestimate the variability of the HK-DTM error.



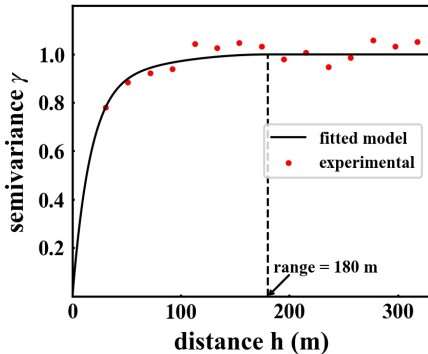

**Figure 4.** Experimental semivariances based on $\epsilon^*_{KK}\{K=180\}$ and fitted parametrized semivariogram model given by Eq. (9). The range of the semivariogram model is $180\,m$. It indicates the maximum autocorrelation length of DEM error, on which the size of the spaial moving filter d depends (see section 3.1).

Based on $\epsilon^*_{KK}\{K=180\}$, we can determine an isotropic semivariogram model which describes the spatial autocorrelation of the HK-DTM error. It results in

$$\gamma(h) = 0.1 \times Sph(\frac{h}{180}) + 0.9 \times Exp(\frac{h}{50}). \tag{9}$$

Here, $Sph(\cdot)$ and $Exp(\cdot)$ denote the basic spherical and exponential semivariogram models (Goovaerts, 1997) and $h$ denotes the horizontal distance between any two locations. A comparison between the experimental semivariance values based on $\epsilon^*_{KK}\{K=180\}$ and the parametrized semivariogram model given by Eq. (9) can be seen in Fig. 4. Semivariance is a measure of spatial dependence between DEM error values at two different locations. The continuous semivariogram model is fitted to the experimental semivariance values so as to deduce semivariance values for any possible distance $h$ required by simple 325 kriging (Goovaerts, 1997). The range of the semivariogram model is $180\,m$. It indicates the maximum autocorrelation length of the HK-DTM error, on which the size of the spatial moving filter d depends (see section 3.1).

### 5.2.2 DEM uncertainty scenarios

As mentioned in section 3, DEM users are often restricted to DEM error information in the form of a single RMSE value per data product. Rarely, they have higher accurate reference data. In order to account for both situations, two corresponding 330 'information levels' are considered in the following study.

A) Rudimentary error information: the RMSE only. In this situation, the RMSE is assumed to be the only available error information of the $5\,m$ resolution HK-DTM. In order to compare results to B), we employ the RMSE $3.3\,m$ as generated based on $\boldsymbol{Z}^*_{KK}\{K=180\}$, as well as the size of the spatial moving filter d $180\,m$ to match the range of the fitted semivariogram model in Fig. 4. USS introduced in section 3.1 is used to generate N realizations of the HK-DTM, 335 denoted as $USS_N\{RMSE=3.3, d=180\}$.





B) Highly informed: higher accurate reference data. In this situation, $\boldsymbol{Z}^*_{KK}\{K = 180\}$ is assumed to be available. That means we know the error $\epsilon^*_{KK}\{K = 180\}$ at the reference locations exactly and the fitted semivariogram model based on $\epsilon^*_{KK}\{K = 180\}$. CSS introduced in section 3.2 is used to generate N realizations of the HK-DTM, denoted as CSS$_N$.

Following the two nominal scenarios A) and B) that are based on specific error $\epsilon^*_{KK}\{K = 180\}$ at reference locations
determined from the available data sources, we also want to analyze the impact of *unrepresentative RMSE* and *subjective d*
issues of USS as introduced in section 3.1 in the form of a sensitivity analysis. Hence, to what extent can we trust the results
of USS if only a single RMSE value per data product is available. Additional three values of the RMSE that are $0.5\,m$, $1.5\,m$,
and $2.5\,m$ with a fixed d $180\,m$ are used as inputs for USS to study the *unrepresentative RMSE* issue. It should be noted that
the RMSE $1.5\,m$ corresponds to the 'true' standard deviation $\sigma$ based on $\epsilon^*_{KK}\{K = 180\}$, see Fig. 3 (b). Another additional
three values of d that are $0\,m$, $90\,m$, and $270\,m$ with a fixed RMSE $3.3\,m$ are used to consider the *subjective d* issue. The
corresponding realizations of the HK-DTM are denoted as USS$_N${RMSE=0.5, 1.5, 2.5, d=180} and USS$_N${RMSE=3.3, d=0,
90, 270}. To sum up, all the scenarios for stochastic simulation are listed in table 1.

**Table 1.** Scenarios for stochastic simulation

| Method | Input ($m$) | Source/Purpose |
|---|---|---|
| | RMSE=3.3, d=180 | $\boldsymbol{Z}^*_{KK}\{K = 180\}$ |
| USS | RMSE=0.5, 1.5, 2.5, d=180 | *unrepresentative RMSE* |
| | RMSE=3.3, d=0, 90, 270 | *subjective d* |
| CSS | semivariogram, $\epsilon^*_{KK}\{K = 180\}$ | $\boldsymbol{Z}^*_{KK}\{K = 180\}$ |

### 5.2.3 Number of DEM realizations

The integrity of a stochastic simulation requires a large number of DEM realizations, while more realizations naturally take
many computational resources. Thus one has to find a reasonable compromise. Typically, this can be found through a represen-
tative convergence study. Since in our study we address the impact of topographic uncertainty on landslide run-out simulation,
we analyze the relative change of topographic attributes with an increasing number of HK-DTM realizations in a preliminary
study. Herein, 1000 HK-DTM realizations are generated for the two 'information levels' A) and B) as introduced in sec-
tion 5.2.2 respectively, namely USS$_{N=1000}${RMSE=3.3, d=180} and CSS$_{N=1000}$. Topographic attributes including slope, aspect,
and ruggedness at all HK-DTM grid points are calculated for each realization.

We define an indicator of the relative change similarly as in Raaflaub and Collins (2006) to investigate the converging
behaviour. Taking slope as an example, for a given number $n$ of HK-DTM realizations, we first calculate the standard deviation
of slope at each HK-DTM gird point over the $n$ realizations. The calculated standard deviation values at all grid points constitute
a grid of standard deviation values. Then we calculate the standard deviation of the grid of standard deviation values, which
leads to a single standard deviation value for the given number $n$. For each $n$ from 1 to 1000, we can correspondingly calculate
a standard deviation value. The same procedure is applied to aspect, ruggedness, and elevation.





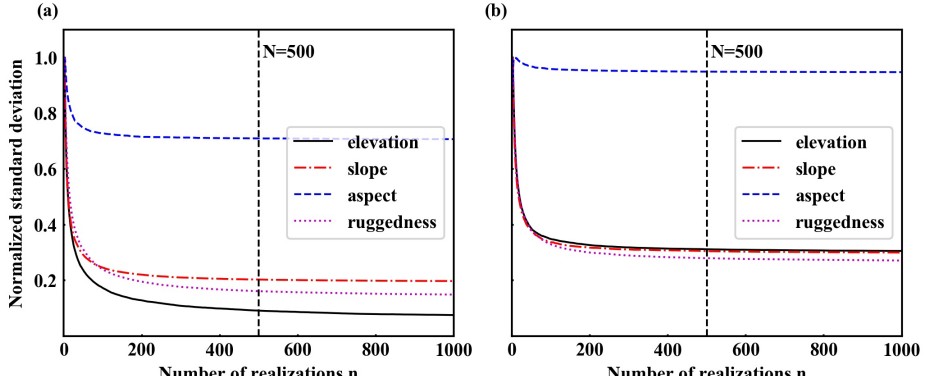

**Figure 5.** The relative change of topographic attributes with respect to the number of HK-DTM realizations. The realizations are generated with (a) $\text{USS}_\text{N}\{\text{RMSE=3.3, d=180}\}$; (b) $\text{CSS}_\text{N}$. Beyond N=500, adding more realizations has little impact on topographic attributes. Therefore, we set N=500 for all computational scenarios in Table 1.

Figure 5 shows plots of normalized standard deviation of the grid of standard deviation values with respect to the number of HK-DTM realizations for the two situations A) and B). It can be seen that for situation A), aspect levels out first, followed by slope, ruggedness, and elevation. Beyond 500 realizations, there is little change of normalized standard deviations. This indicates adding more realizations has little impact on topographic attributes. For situation B), aspect also levels out first while the rest three show less difference. Compared to A), B) converges faster which indicates CSS requires less DEM realizations than USS. Nevertheless, we set N=500 for the remainder of this study both for USS and CSS. Namely, we generate 500 HK-DTM realizations for each scenario input set as listed in Table 1.

### 5.2.4 Statistics of DEM error realizations

In order to conduct a further quality check of our implementation of both USS and CSS, we investigate the corresponding DEM error realizations of the $\text{USS}_\text{N=500}\{\text{RMSE=3.3, d=180}\}$ and $\text{CSS}_\text{N=500}$ scenarios, denoted as $\text{USS}_\text{N=500}^\text{Error}\{\text{RMSE=3.3, d=180}\}$ and $\text{CSS}_\text{N=500}^\text{Error}$ respectively. Ideally, the local mean $\mu_{ij}$ and standard deviation $\sigma_{ij}$ of DEM error realizations at each grid point $D_{ij}$ should match the underlying assumptions as introduced in section 3 if the number of DEM error realizations is sufficiently large.

Figure 6 (a) and (c) show the mean and standard deviation grid of the $\text{USS}_\text{N=500}^\text{Error}\{\text{RMSE=3.3, d=180}\}$. It can be seen that the mean values at all grid points are centered around $0\,m$. The standard deviation values are centered around $3.3\,m$. This corresponds to the assumption underlying USS that all $\epsilon_{ij}$ fulfill a same univariate Gaussian distribution with a mean ($\mu$) of zero and a standard deviation ($\sigma$) given by the RMSE (see section 3.1).

Figure 6 (b) and (d) show the mean and standard deviation grid of the $\text{CSS}_\text{N=500}^\text{Error}$. The mean values at grid points away from the reference locations are centered around the mean ($\mu$) $-3.0\,m$ based on $\epsilon_{KK}^*\{K=180\}$. They become close to $\epsilon_{KK}^*\{K=180\}$ with the decrease of distance between grid points and the reference locations, and are equal to $\epsilon_{KK}^*\{K=180\}$ at the

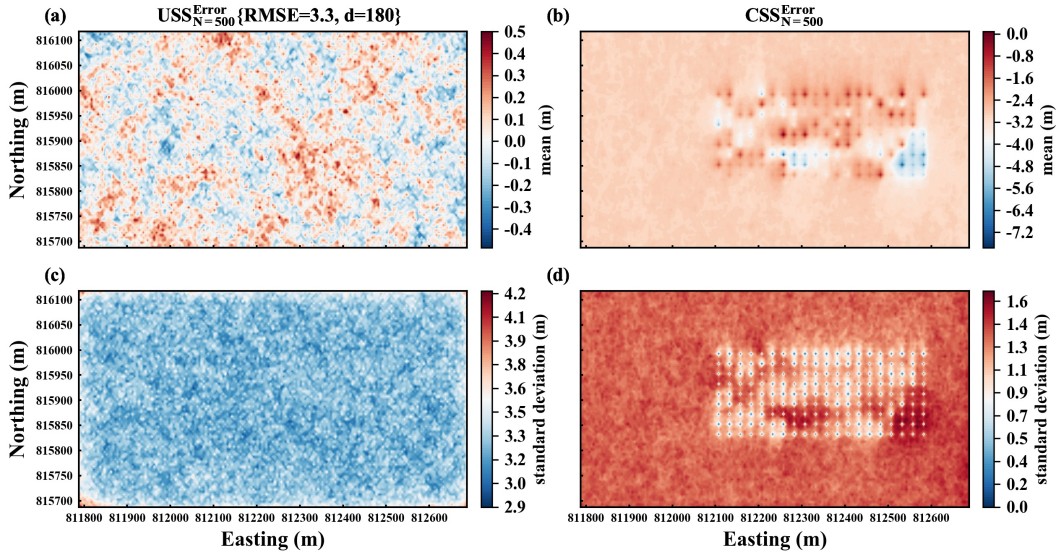

**Figure 6.** Statistics of HK-DTM error realizations. (a) mean and (c) standard deviation grid of $\mathrm{USS}_{N=500}^{\mathrm{Error}}\{\mathrm{RMSE}=3.3, \mathrm{d}=180\}$. The mean and standard deviation values are centered around $0\,m$ and $3.3\,m$; (b) mean and (d) standard deviation grid of $\mathrm{CSS}_{N=500}^{\mathrm{Error}}$. The mean values at grid points away from the reference locations are centered around the mean ($\mu$) $-3.0\,m$ of $\epsilon_{KK}^*\{K=180\}$ and are equal to $\epsilon_{KK}^*\{K=180\}$ at the reference locations. The standard deviation values at grid points away from the reference locations are centered around the standard deviation ($\sigma$) $1.5\,m$ of $\epsilon_{KK}^*\{K=180\}$ and vanish at the reference locations. This matches the assumptions underlying USS and CSS as introduced in section 3.

reference locations. Similarly, the standard deviation values at grid points away from the reference locations are centered around the standard deviation ($\sigma$) $1.5\,m$ based on $\epsilon_{KK}^*\{K=180\}$. They vanish at the reference locations. This also corresponds to the assumption underlying CSS that each $\epsilon_{ij}$ fulfill a univariate Gaussian distribution with a mean $\mu_{ij}$ and standard deviation $\sigma_{ij}$ given by the simple kriging estimate and simple kriging standard deviation at $D_{ij}$ (see section 3.2).

### 5.3 Landslide process simulation setup

With the DEM realizations generated in section 5.2, we can study the impact of DEM uncertainty on landslide process simulation. Here, we introduce the key inputs and our setup for the process simulation.

#### 5.3.1 Model input

Release zone area and fracture height, friction parameters, and a DEM are three key inputs for performing a deterministic landslide process simulation based on the VS model and utilizing the mass movement simulation platform RAMMS (Christen et al., 2010). For all scenarios, we consistently use the release zone area as provided for the benchmark exercise during the second JTC1 workshop, which match that of the 2008 Yu Tung Road landslide (Pastor et al., 2018) as shown in Fig. 7 (b).





The fracture height is assumed to be $1.2\,m$ leading to a release volume of around $2900\,m^3$ based on the $5\,m$ resolution HK-
DTM. The friction parameters $\mu$ and $\xi$ used in this study are 0.105 and $300\,m/s^2$ respectively. They are suggested in the GEO
report issued by Civil Engineering and Development Department of Hong Kong, which are obtained using back-analysis with
information from a video capturing the lower portion of the landslide and detailed field mapping after the landslide (AECOM
Asia Company Limited, 2012). The HK-DTM and all HK-DTM realizations generated in section 5.2 are used as DEM inputs.
Entrainment is not considered in this study.

### 5.3.2 Simulation ensembles

We denote a deterministic landslide process simulation based on a DEM as a simulation run and N deterministic landslide
process simulations based on N DEM realizations as a simulation ensemble. The following deterministic simulation and simu-
lation ensembles are conducted based on the original HK-DTM and the aforementioned computational scenarios, see Table 1.
They are named after the corresponding DEM and DEM realizations.

1) Deterministic simulation HK-DTM: one landslide process simulation run is conducted based on the original HK-DTM.
    This one time simulation corresponds to, what is traditionally done in a simulation based hazard assessment study. The
    results serve as the basis to assess the impact of DEM uncertainty.

   2) $\text{USS}_{N=500}\{\text{RMSE}=3.3, \text{d}=180\}$ ensemble: 500 landslide process simulations are conducted based on the $\text{USS}_{N=500}\{\text{RMSE}=3.3, \text{d}=180\}$ DEM realizations as introduced in section 5.2. Each of them is referred to as $\text{USS}^n_{N=500}\{\text{RMSE}=3.3,$
d=180\}$, with $n=1, 2, ..., 500$. This ensemble allows us to access the impact of DEM uncertainty if only the RMSE is
    available.

   3) $\text{CSS}_{N=500}$ ensemble: 500 landslide process simulations are conducted based on the $\text{CSS}_{N=500}$ DEM realizations. Similar
    to 2), each of them is referred to as $\text{CSS}^n_{N=500}$ with $n=1, 2, ..., 500$. This ensemble allows us to assess the impact of DEM
    uncertainty if higher accurate reference data is available.

4) $\text{USS}_{N=500}\{\text{RMSE}=0.5, 1.5, 2.5, \text{d}=180\}$ ensembles: 500 landslide process simulations are conducted for three different
    RMSE values respectively while keeping the maximum autocorrelation length d constant. They lead to 1500 process
    simulations. The results allow us to discuss the *unrepresentative RMSE* issue as detailed in section 3.1. They can be also
    used to discuss the relationship between the degree of DEM uncertainty and its impact.

   5) $\text{USS}_{N=500}\{\text{RMSE}=3.3, \text{d}=0, 90, 270\}$ ensembles: 500 landslide process simulations are conducted for three different
maximum autocorrelation length values respectively while keeping the RMSE constant. They lead to 1500 process
    simulations. The results allow us to discuss the *subject d* issue as detailed in section 3.1.

All in all this adds up to one deterministic simulation run HK-DTM, as well as to simulation ensembles 500 process simulations
each, the $\text{USS}_{N=500}\{\text{RMSE}=3.3, \text{d}=180\}$ ensemble and $\text{CSS}_{N=500}$ ensemble, that are constructed from higher accurate reference
data based on the 2m-DEM, as well as 3000 additional process simulations to result in six ensembles $\text{USS}_{N=500}\{\text{RMSE}=0.5,$



1.5, 2.5, d=180} and USS$_{N=500}${RMSE=3.3, d=0, 90, 270} to test sensitivities. Each process simulation takes around one minute on a laptop with Intel Core i7-9750H CPU, adding up to around 67 hours simulation time.

## 6 Results and discussions

This section is organized according to the simulation ensembles introduced in section 5.3.2. Section 6.1 presents the results of the deterministic simulation HK-DTM which serves as the basis for all following discussions. Section 6.2 is devoted to analyze

the impact of DEM uncertainty on landslide process simulation in the case of RMSE only (USS$_{N=500}${RMSE=3.3, d=180} ensemble) and available higher accurate reference data (CSS$_{N=500}$ ensemble). In section 6.3, the *unrepresentative RMSE* and *subjective d* issues are discussed based on the ensembles described in section 5.3.2 4) and 5).

### 6.1 Deterministic simulation HK-DTM

In a continuum mechanical landslide process model such as used for this study and introduced in section 2, the landslide

flow behaviour is characterised by its spatially varying height and velocity distribution over time, denoted as $H(x, y, t)$ and $\boldsymbol{U}(x, y, t)$. For the purpose of landslide hazard assessment and mitigation measure development, hence maximum height and velocity data through the duration of the landslide are most informative. Thus, we focus on the maximum values of $H(x, y, t)$ and $\boldsymbol{U}(x, y, t)$ over all time, denoted as $H_{\max}(x, y)$ and $\|\boldsymbol{U}_{\max}(x, y)\|$.

Figure 7 (a) and (b) show $H_{\max}(x, y)$ and $\|\boldsymbol{U}_{\max}(x, y)\|$ as given by the deterministic simulation HK-DTM. It should be noted

that there is a relatively high elevation area at the end part of the channel in the HK-DTM as denoted within the red circle in Fig. 7 (b). It corresponds to the construction of debris-resisting barriers after the 2008 Yu Tung Road landslide as introduced in section 5.1. The flow material is decelerated and held back here. We will come back to this point latter in section 6.2.1.

Landslide run-out distance is often characterised in terms of its apparent friction angle. The tangent of the apparent friction angle is equal to the ratio of the landslide fall height and the run-out distance (DeBlasio and Elverhoi, 2008). The apparent

friction angle evaluated from the deterministic simulation is 16.80°. These results are used as reference to assess the impact of DEM uncertainty in the following discussions.

### 6.2 USS$_{N=500}${RMSE=3.3, d=180} ensemble and CSS$_{N=500}$ ensemble

While it is straightforward to present results of a deterministic simulation run as shown in section 6.1, a stochastic simulation based ensembles of N simulation run call for tailored statistic to manage and interpret the extensive output data. First, we define

the hazard probability $P_{(x_l, y_l)}$ at one location $(x_l, y_l)$ as the frequency of $H_{\max}(x_l, y_l)$ exceeding a certain pre-defined height threshold value, hence

$$P_{(x_l, y_l)} = \frac{\sum_{n=1}^{N} P_{(x_l, y_l)}^n}{N}, \quad P_{(x_l, y_l)}^n = \begin{cases} 1, & \text{if } H_{\max}^n(x_l, y_l) \geq \text{threshold} \\ 0, & \text{otherwise} \end{cases} \tag{10}$$

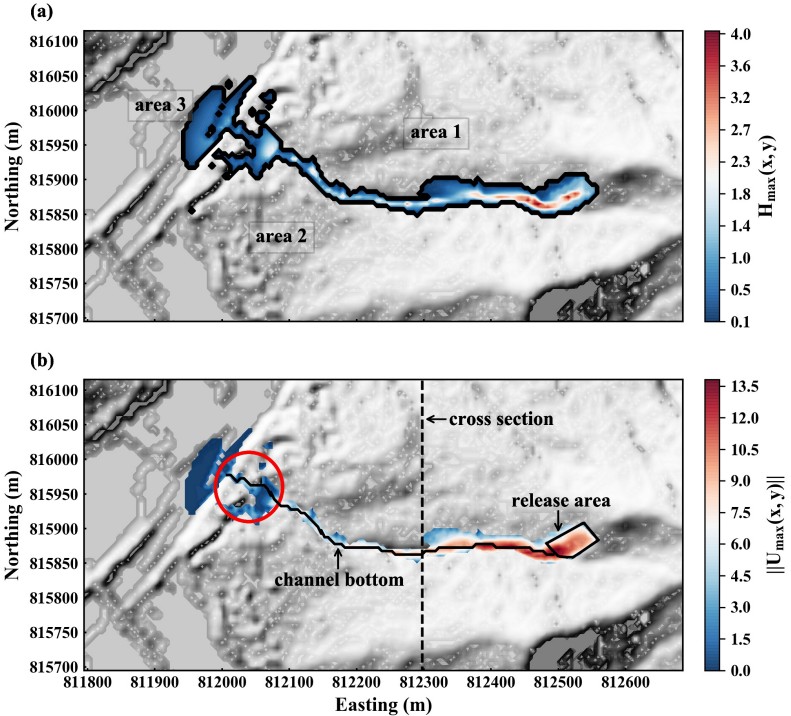

**Figure 7.** Results of the deterministic simulation HK-DTM. (a) $H_{\max}(x, y)$ above a cut-off threshold of $0.1\,m$. The black outline is the $0.1\,m$ contour of $H_{\max}(x, y)$. The area within this outline is regarded as hazard area. Area 1-3 are denoted for latter discussions. (b) $\|\boldsymbol{U}_{\max}(x, y)\|$ above a cut-off threshold of $0.01\,m/s$. The relatively high elevation area within the red circle decelerates and holds back the flow material. The channel bottom and cross section are denoted for latter discussions.

where $H_{\max}^n(x_l, y_l)$ denotes the maximum flow height at location $(x_l, y_l)$ for the $n$-th simulation run of the corresponding ensemble. $P_{(x_l, y_l)}^n$ hence informs whether location $(x_l, y_l)$ is within the hazard area of the $n$-th simulation run for a given

threshold, and $P_{(x_l, y_l)}$ about the resulting hazard probability at location $(x_l, y_l)$ considering the complete ensemble. Here, the threshold is set as $0.1\,m$ which matches the cut-off threshold of the deterministic simulation HK-DTM in Fig. 7 (a). Evaluation of hazard probabilities at all locations then gives rise to a probabilistic hazard map (Stefanescu et al., 2012), which provides an overall view of the DEM uncertainty impact.

Besides assessing the overall impact of DEM uncertainty in terms of the probabilistic hazard map, we will also discuss the

local impact of DEM uncertainty on dynamic flow properties, focusing on $H_{\max}(x, y)$ and $\|\boldsymbol{U}_{\max}(x, y)\|$ at locations along the channel bottom and the channel cross section denoted in Fig. 7 (b).

### 6.2.1   Probabilistic hazard maps

Figure 8 (a) and (c) show the probabilistic hazard map for both USS$_{N=500}$ {RMSE=3.3, d=180} ensemble and CSS$_{N=500}$ ensemble. It can be seen that the potential hazard area is much larger than the deterministic hazard area for both ensembles. The




difference between the deterministic and the ensemble-based hazard area is most pronounced in area 1-3 for USS$_{N=500}$ {RMSE= 3.3, d=180} ensemble and in area 3 for CSS$_{N=500}$ ensemble. Fig. 8 (b) and (d) show boxplots of the apparent friction angle distribution for both ensembles. It is evident that the apparent friction angle of both ensembles varies largely with respect to the apparent friction angle of the deterministic simulation (16.80°). CSS$_{N=500}$ ensemble-based apparent friction angle (mean 15.39°) is smaller than USS$_{N=500}$ {RMSE=3.3, d=180} ensemble-based apparent friction angle (mean 16.76°). This can be
explained as follows.

Due to DEM uncertainty, topographic characteristics represented in DEM realizations vary from that represented in the original DEM. Specifically: 1) topographic details of the deterministic channel tend to be dampened out from DEM realizations. The topographic details include banks of the channel, as well as relatively high elevation area at the end part of the channel that holds back flow material as shown in Fig. 7 (b); 2) topographic roughness tends to increase.

Whether, where, and to what extent the topographic characteristics in DEM realizations would differ from the original DEM depend on: 1) variability of DEM error. Intuitively, the larger the variability, the more likely that topographic details of the deterministic channel would be dampened out, and the larger the topographic roughness in DEM realizations; 2) topographic details of the original DEM. If subject to the same DEM error, less 'well defined' topographic characteristics in the original DEM are more likely to be changed in DEM realizations. For example, along the channel of the HK-DTM, the north bank of
the channel near area 1 and the south bank of the channel near area 2 are less 'well defined' compared to other parts of channel banks. Flow material could be more easily diverted to area 1 and area 2 where elevations are relatively low and some local 'small channels' exist. Area 3 could also be regarded as less 'well defined' since it is relatively flat and thus is sensitive to DEM uncertainty (Temme et al., 2009).

The change of each topographic characteristic has corresponding impact on landslide run-out behaviour. Specifically: 1) if
banks of the deterministic channel were dampened out in DEM realizations, flow material tends to spread out along channel cross section direction and travel distance would be shorter; 2) if the relatively high elevation area that holds back flow material was dampened out, flow material tends to travel further; 3) increasing topographic roughness leads to higher simulated momentum losses and shorter travel distance as pointed out by McDougall (2017).

For USS$_{N=500}$ {RMSE=3.3, d=180} ensemble, the variability of DEM error is relatively large, e.g. around $3.3\,m$ governed by
the not-bias-corrected RMSE based on $\epsilon^*_{KK}\{K=180\}$ (see Fig. 6 (c)). In this situation, both the north bank near area 1 and south bank near area 2 as well as the relatively high elevation area at the end part of the channel are possible to be dampened out in HK-DTM realizations. For CSS$_{N=500}$ ensemble, the variability of DEM error is relatively small, e.g. around $1.5\,m$ governed by the standard deviation ($\sigma$) based on $\epsilon^*_{KK}\{K=180\}$ (see Fig. 6 (d)). In this situation, the banks tend to remain 'well defined' while the relatively high elevation area is possible to be dampened out in HK-DTM realizations. Thus, area 1 and area 2 are
possibly subject to hazard in USS$_{N=500}$ {RMSE=3.3, d=180} ensemble but less likely in CSS$_{N=500}$ ensemble. As mentioned above, area 3 is a flat area which is sensitive to DEM uncertainty. Furthermore, it locates near the deposition, around which the impact of upstream DEM uncertainty seems to accumulate. Thus, it is highly affected in both ensembles.

The apparent friction angle distribution is determined by a combined effect of change of channel banks, change of the relatively high elevation area at the end part of the channel, and increasing topographic roughness. For USS$_{N=500}$ {RMSE=3.3,


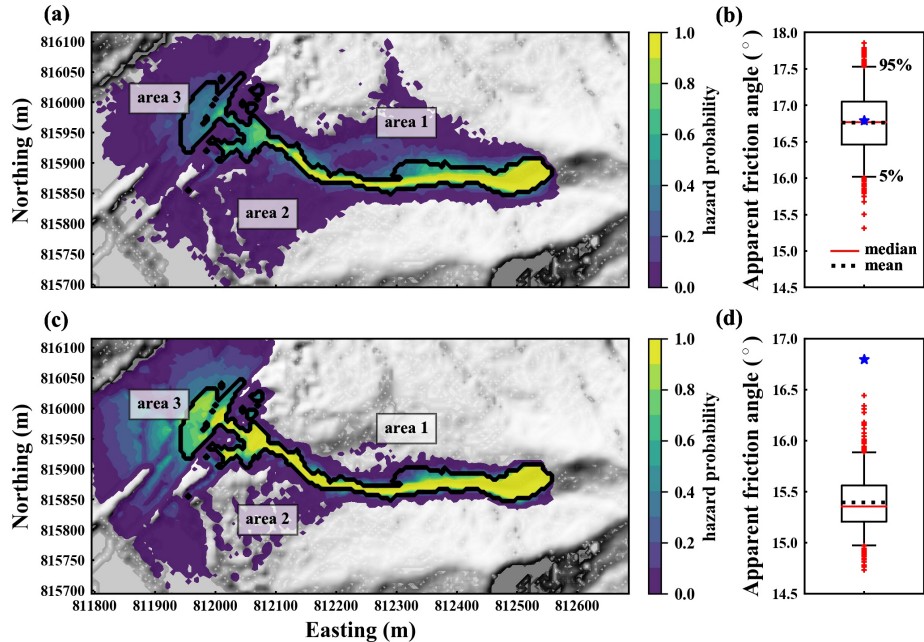

**Figure 8.** (a) Probabilistic hazard map and (b) corresponding apparent friction angle distribution of USS$_{N=500}$ {RMSE=3.3, d=180} ensemble; (c) probabilistic hazard map and (d) corresponding apparent friction angle distribution of CSS$_{N=500}$ ensemble. The black outline plotted on the hazard maps represents the deterministic hazard area (see Fig. 7 (a)). In the boxplots, the blue star denotes the apparent friction angle of the deterministic simulation HK-DTM (see section 6.1). The difference between the deterministic and the ensemble based hazard area is most pronounced in area 1-3 for USS$_{N=500}$ {RMSE=3.3, d=180} ensemble and in area 3 for CSS$_{N=500}$ ensemble. Our main findings are: 1) accounting for DEM uncertainty may significantly increase the potential hazard area; 2) the potential hazard area is highly related to the variability of DEM error and topographic details of the original DEM; 3) USS based on the RMSE only may overestimate the spread of potential hazard area and underestimate the travel distance due to an *unrepresentative RMSE* (e.g. not bias-corrected) that overestimates the variability of DEM error.

d=180} ensemble, deteriorated channel bank representation and increasing topographic roughness make flow material travel shorter distance, e.g. larger apparent friction angle, while deteriorated relatively high elevation area representation allows flow material to travel further, e.g. smaller apparent friction angle. For CSS$_{N=500}$ ensemble, channel banks are likely to remain 'well defined' and the degree of topographic roughness increase is lower due to its relatively small variability of DEM error compared to USS$_{N=500}$ {RMSE=3.3, d=180} ensemble. Thus, flow material in CSS$_{N=500}$ ensemble tends to travel longer distance, e.g. smaller apparent friction angle, compared to USS$_{N=500}$ {RMSE=3.3, d=180} ensemble.


In summary, we can conclude from the probabilistic hazard maps and boxplots of apparent friction angle distribution that: 1) accounting for DEM uncertainty may significantly increase the potential hazard area; 2) the potential hazard area is highly related to the variability of DEM error and topographic characteristics of the original DEM; 3) USS based on the RMSE only





may overestimate the spread of potential hazard area and underestimate travel distance due to a not-bias-corrected RMSE that
overestimates the variability of DEM error.

It should be noted that the probabilistic hazard map here is constructed based on maximum height and a pre-defined threshold. In simulation-based hazard assessment practice, one may indicate potential hazard using other indicators, e.g. maximum momentum that reflects the impact pressure, etc. and correspondingly modify the threshold value. In this case, our workflow is easily extendible to account for other indicators.

**6.2.2 Dynamic flow properties**

The left column in Fig. 9 shows elevation, maximum height and maximum velocity at locations along the channel bottom based on $USS_{N=500}$ {RMSE=3.3, d=180} ensemble. It is evident that both maximum height and maximum velocity at these locations largely vary from that of the deterministic simulation. Specifically, the mean of maximum height (maximum velocity) values at all the locations based on the deterministic simulation is $1.28\,m$ ($7.17\,m/s$). The mean of ensemble-based 90% confidence
interval of maximum height (maximum velocity) is $[0.18\,m, 2.17\,m]$ ($[0.99\,m/s, 7.89\,m/s]$) (e.g. the range between the mean of ensemble-based 5% percentile and the mean of ensemble-based 95% percentile). Another observation is that ensemble-based mean of flow dynamic properties is generally smaller than the mean of flow dynamic properties of the deterministic simulation (e.g. the red dashed line is generally under the black line in both Fig. 9 (c) and (e)). The mean of ensemble-based mean of maximum height (maximum velocity) is $0.85\,m$ ($4.57\,m/s$), around 66% (64%) of the mean of the deterministic simulation
$1.28\,m$ ($7.17\,m/s$) (see Fig. 9 (c) and (e)).

The right column in Fig. 9 shows corresponding results based on $CSS_{N=500}$ ensemble. Similar trends as in $USS_{N=500}$ {RMSE=3.3, d=180} ensemble can also be observed. Namely, both maximum height and maximum velocity at these locations largely vary from that of the deterministic simulation, and ensemble-based mean of flow dynamic properties is generally smaller than deterministic results. Main differences are that the variation range of $CSS_{N=500}$ ensemble-based flow dynamic properties is
smaller, and $CSS_{N=500}$ ensemble-based mean of flow dynamic properties is larger compared to $USS_{N=500}$ {RMSE=3.3, d=180} ensemble. More specifically, the mean of $CSS_{N=500}$ ensemble-based 90% confidence interval of maximum height (maximum velocity) is $[0.5\,m, 2.03\,m]$ ($[3.56\,m/s, 7.99\,m/s]$). The mean of $CSS_{N=500}$ ensemble-based mean of maximum height (maximum velocity) is $1.1\,m$ ($6.01\,m/s$), around 86% (84%) of the mean of the deterministic simulation $1.28\,m$ ($7.17\,m/s$) (see Fig. 9 (d) and (f)).

The above observations result from similar factors as discussed in section 6.2.1. Due to DEM uncertainty,

- ensemble-based flow dynamic properties are likely to vary from that of the deterministic simulation. Larger variability of DEM error is likely to result in more extreme results. As discussed in section 6.2.1, the variability of DEM error for $USS_{N=500}$ {RMSE=3.3, d=180} ensemble is larger than that for $CSS_{N=500}$ ensemble due to *unrepresentative RMSE* issue. Thus the variation range of $USS_{N=500}$ {RMSE=3.3, d=180} ensemble-based flow dynamic properties is generally larger
than that of $CSS_{N=500}$ ensemble-based flow dynamic properties, e.g. larger mean of ensemble-based 90% confidence interval (the trend would be more clear if we also consider outliers outside 90% confidence interval).


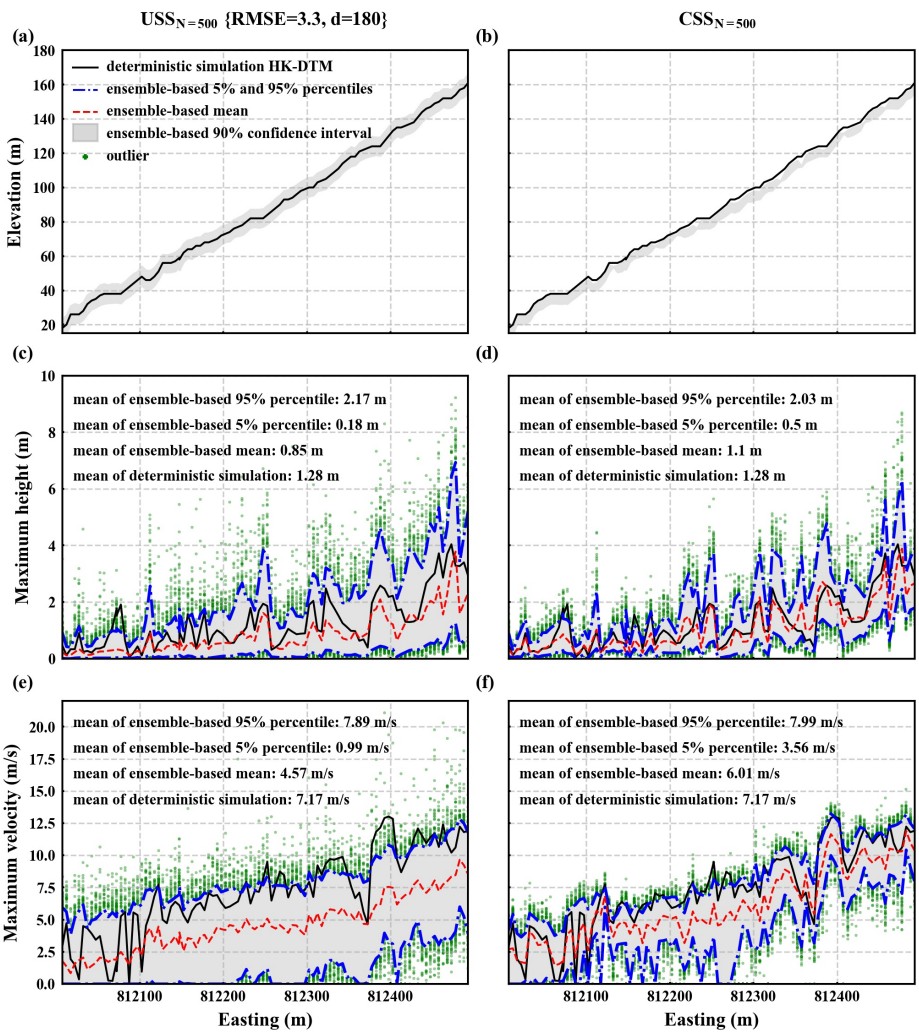

**Figure 9.** Elevation, maximum height, and maximum velocity at locations along the channel bottom (see Fig. 7 (b)). Left and right columns correspond to USS$_{N=500}$ {RMSE=3.3, d=180} ensemble and CSS$_{N=500}$ ensemble respectively. In each subfigure, blue dashed dotted lines represent ensemble-based 5% and 95% percentiles of the quantity. The red dashed line represents ensemble-based mean of the quantity. The black line denotes corresponding results of the deterministic simulation. Annotated mean values are average of all the locations. Ensemble-based flow dynamic properties largely vary from deterministic simulation results. The variation range of USS$_{N=500}$ {RMSE=3.3, d=180} ensemble is larger while its ensemble-based mean is smaller, compared to counterparts of CSS$_{N=500}$ ensemble. Our main findings are: 1) accounting for DEM uncertainty may significantly affect dynamic flow properties hence any hazard assessment that is based on landslide dynamics; 2) USS based on the RMSE only may overestimate the range of dynamic flow properties and underestimate ensemble-based mean of dynamic flow properties due to an *unrepresentative RMSE* that overestimates the variability of DEM error.

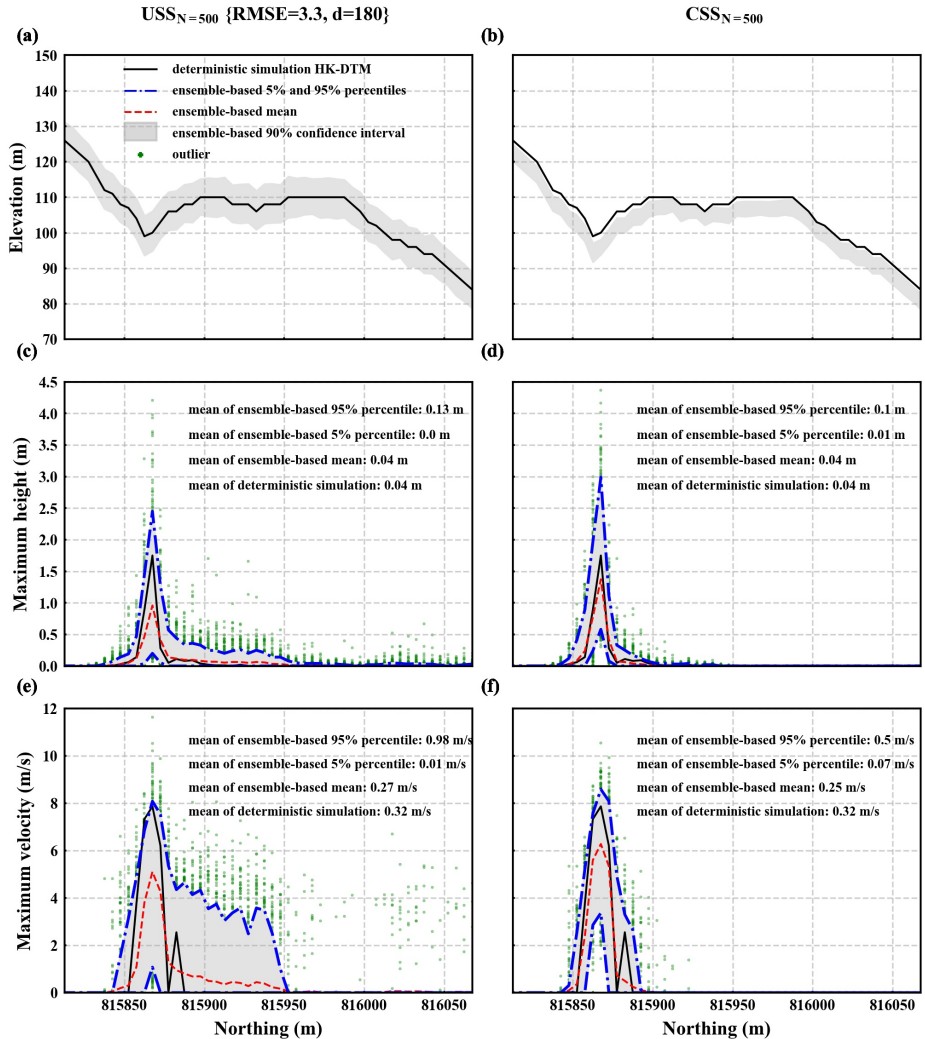

**Figure 10.** Elevation, maximum height, and maximum velocity at locations along the channel cross section (see Fig. 7 (b)). Left and right columns correspond to USS$_{N=500}$ {RMSE=3.3, d=180} ensemble and CSS$_{N=500}$ ensemble respectively. In each subfigure, blue dashed dotted lines represent ensemble-based 5% and 95% percentiles of the quantity. The red dashed line represents ensemble-based mean of the quantity. The black line denotes corresponding results of the deterministic simulation. Annotated mean values are average of all the locations. Due to DEM uncertainty, flow material of both ensembles tends to spread out along the channel cross section direction. The ensemble-based mean of flow dynamic properties at the channel bottom location is smaller than flow dynamic properties at the channel bottom location of the deterministic simulation (compare peak value of red dashed line with peak value of black line). The more the flow material spreads out, the smaller the ensemble-based mean of flow dynamic properties at the channel bottom location (compare results of USS$_{N=500}$ {RMSE=3.3, d=180} ensemble with that of CSS$_{N=500}$ ensemble).




- banks of the deterministic channel may be dampened out in DEM realizations. Deteriorated channel bank representation makes flow material more spread out along channel cross section direction. This could lead to smaller ensemble-based mean of flow dynamic properties at channel bottom locations, compared to flow dynamic properties of the deterministic simulation. It can be directly seen in Fig. 10, which displays results of one channel cross section. Also, due to larger variability of DEM error, flow material in $USS_{N=500}$ {RMSE=3.3, d=180} ensemble is more spread along channel cross section direction, resulting in smaller ensemble-based mean of flow dynamic properties at channel bottom locations compared to $CSS_{N=500}$ ensemble. This can also be seen in Fig. 10.

- topographic roughness in DEM realizations tends to increase. As pointed out in section 6.2.1, increasing topographic roughness results in higher simulated momentum losses and thus smaller flow dynamic properties on average. The higher the degree of topographic roughness increase, the higher the simulated momentum losses and the smaller the flow dynamic properties. This also contributes to smaller ensemble-based mean of flow dynamic properties at channel bottom locations, compared to flow dynamic properties of the deterministic simulation, as well as to smaller $USS_{N=500}$ {RMSE=3.3, d=180} ensemble-based mean of flow dynamic properties at channel bottom locations, compared to $CSS_{N=500}$ ensemble.

Based on the ensembles' dynamic flow properties we can conclude that: 1) accounting for DEM uncertainty may significantly affect dynamic flow properties, e.g. maximum height and maximum velocity, hence any hazard assessment that is based on landslide dynamics; 2) USS based on the RMSE only may overestimate the range of dynamic flow properties and underestimate ensemble-based mean of dynamic flow properties due to an *unrepresentative RMSE* that overestimates the variability of DEM error.

## 6.3 Additional ensembles to investigate USS sensitivities in RMSE and d

Here, we discuss the *unrepresentative RMSE* and *subjective d* issues as introduced in section 3.1 based on additional six ensembles $USS_{N=500}$ {RMSE=0.5, 1.5, 2.5, d=180} and {RMSE=3.3, d=0, 90, 270} (refer to section 5.3.2) as well as the $USS_{N=500}$ {RMSE=3.3, d=180} ensemble. Results of the $CSS_{N=500}$ ensemble are used as a reference since $CSS_{N=500}$ incorporated more information on the DEM error. It is thus reasonable to assume that its results reflect the reality better.

Figure 11 shows the consolidated results of the ensembles. The left, middle, and right column correspond to set of $USS_{N=500}$ {RMSE=0.5, 1.5, 2.5, 3.3, d=180} ensembles, set of $USS_{N=500}$ {RMSE=3.3, d=0, 90, 180, 270} ensembles, and $CSS_{N=500}$ ensemble respectively. The first row shows stacked bar plots of the potential hazard area's magnitude based on the probabilistic hazard map for each ensemble (see Fig. 8 (a) and (c)). The second row shows apparent friction angle distribution. The last two rows show statistics of maximum height and maximum velocity at channel bottom locations (see Fig. 9 (c)-(f)). Also, deterministic simulation results are included.

Focusing on the left column, it can be seen that with increasing RMSE: 1) low-probability (0-0.2) hazard area significantly increases and high-probability (0.8-1) hazard area gradually decreases leading to increase of overall potential hazard area if we keep the same threshold value; 2) except for the RMSE=0.5 $m$ ensemble, the apparent friction angle steadily increases; 3)


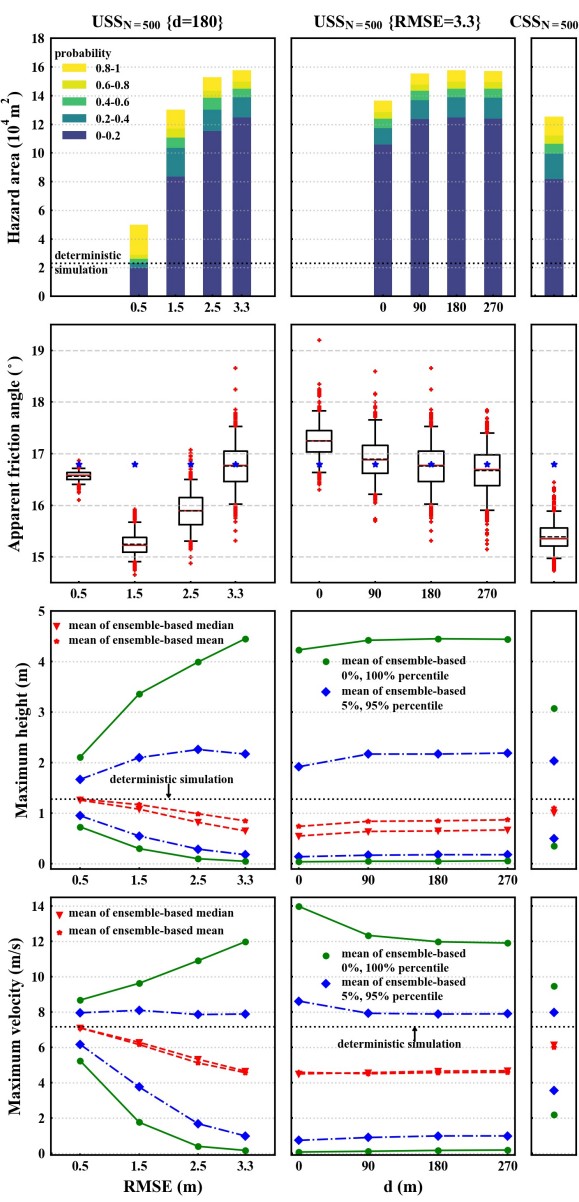

**Figure 11.** Consolidated results of all ensembles. The left, middle, and right columns correspond to set of $USS_{N=500}$ {RMSE=0.5, 1.5, 2.5, 3.3, d=180} ensembles, set of $USS_{N=500}$ {RMSE=3.3, d=0, 90, 180, 270} ensembles, and $CSS_{N=500}$ ensemble respectively. The first row shows stacked bar plots of the potential hazard area's magnitude based on the probabilistic hazard map for each ensemble (see Fig. 8 (a) and (c)). The second row shows apparent friction angle distribution. The last two rows show statistics of maximum height and maximum velocity at channel bottom locations (see Fig. 9 (c)-(f)).





the range of extreme values of maximum height (maximum velocity) at channel bottom locations increases while average of maximum height (maximum velocity) at channel bottom locations decreases.

For purely RMSE-based USS, the standard deviation of DEM error is assumed to be determined by the RMSE. Hence larger RMSE indicates larger variability of DEM error in DEM realizations. The larger the variability of DEM error, the more likely topographic details of the deterministic channel would be dampened out, and the larger the topographic roughness in DEM

realizations. As discussed in section 6.2.1, this would make flow material more spread out along channel cross section direction (namely larger potential hazard area) and travel shorter distance (namely larger apparent friction angle). As discussed in section 6.2.2, larger variability of DEM error is likely to result in more extreme values of flow dynamic properties (namely larger range of extreme values) while spreading of flow material along channel cross section direction and larger topographic roughness lead to smaller ensemble-based mean of flow dynamic properties at channel bottom locations.

As discussed in section 6.2.1, the apparent friction angle distribution is determined by a combined effect of change of channel banks, change of the relatively high elevation area at the end part of the channel, and increasing topographic roughness. It naturally follows that for a small variability of DEM error (here RMSE=0.5 $m$), all the changes are less significant in DEM realizations and thus the apparent friction angle of USS$_{N=500}${RMSE=0.5, d=180} ensemble closely matches the deterministic simulation result. For an intermediate variability of DEM error (here RMSE=1.5 $m$), the relatively high elevation area at the

end part of the channel is subject to change while channel banks tend to remain 'well defined' in DEM realizations. This leads to longer travel distance of USS$_{N=500}${RMSE=1.5, d=180} ensemble (namely smaller apparent friction angle) in comparison to the deterministic simulation result.

From the middle column of Fig. 11, we find that consistently the results for a USS ensemble of vanishing spatial auto-correlation USS$_{N=500}$ {RMSE=3.3, d=0} differ significantly from USS ensembles that include spatial autocorrelation, hence

USS$_{N=500}${RMSE=3.3, d=90, 180, 270} ensembles. This indicates that whether spatial autocorrelation is considered or not may make a difference but the extent of spatial autocorrelation has less influence on simulation results. As we know spatial autocorrelation to be present in topographic data but often lack information on its exact autocorrelation length, this is actually good news for practical hazard assessment studies.

Comparing the left column of Fig. 11 with the right column, it can furthermore be seen that the results of the USS$_{N=500}$

{RMSE=1.5, d=180} ensemble are quite close to the results of the CSS$_{N=500}$ ensemble. The USS$_{N=500}$ {RMSE=1.5, d=180} ensemble is informed with the bias-corrected RMSE (namely the true standard deviation, in our case 1.5 $m$, see Fig. 3 (b)). It indicates that if a bias-corrected RMSE is given, USS is possible to provide reasonable results considering the extent of spatial autocorrelation has less influence on simulation results.

All in all, we find that: 1) the results of USS are in general more sensitive to values of the RMSE and less sensitive to values

of d; 2) an *unrepresentative RMSE* that overestimates the variability of DEM error may overestimate the potential hazard area and extreme values of dynamic flow properties; 3) whether or not spatial autocorrelation of DEM error is considered can make a difference of ensemble-based simulation results; 4) if a bias-corrected RMSE is given, it is possible to obtain reasonable ensemble-based simulation results using USS.





## 7 Conclusions

In this paper, we investigated different approaches to study the impact of topographic uncertainty on simulation-based flow-like landslide run-out analyses. Based upon a historic landslide event for which two DEM data sets of different accuracy had been available, we presented a series of computational scenarios. Unconditional and conditional stochastic simulation are conducted to generate DEM realizations, both for the case in which only the RMSE is available, and for the case in which reference data of higher accuracy is available. The computational workflow including the stochastic simulation to generate the

DEM realizations and the landslide run-out simulation is implemented as a modular Python-based package. How topographic uncertainty propagates into results of landslide run-out analysis is discussed in detail. In addition, we addressed the two major issues of purely RMSE-based unconditional stochastic simulation, e.g. the fact that not-bias-corrected RMSE overestimates the variability of DEM error (referred to as *unrepresentative RMSE* in our study) and the fact that determining the size of the spatially moving filter in the absence of further information on the spatial DEM error structure is often subjective (referred to

as *subjective d* in our study). Our main conclusions are:

- DEM uncertainty significantly affects simulation-based landslide run-out modeling depending on how well the underlying flow path is represented, which is determined by topographic characteristics of the original DEM and the variability of DEM error. For the same degree of variability of DEM error, the less 'well defined' parts of the flow path in the original DEM are more likely to be affected and thus leads to change of flow behaviour at these parts. Also, an increasing

variability of DEM error leads to an increased impact. More specifically, with increasing variability of the DEM error, the potential hazard area and extreme values of dynamic flow properties are likely to increase. This shows the importance of considering topography induced uncertainty for simulation-based landslide hazard assessment rather than simply trusting results of a deterministic simulation if a high accuracy of DEM source is not guaranteed. Also, a preliminary terrain analysis may give some hints on areas that will potentially be affected by a topographic uncertainty study.

- Both unconditional and conditional stochastic simulation methods can be applied to study DEM uncertainty propagation in landslide run-out modeling. Their main difference is that the computationally performant unconditional stochastic simulation can be conducted based on RMSE information only, while the computationally costly conditional stochastic simulation requires the availability of higher accurate reference data and is thus more accurate. The higher accurate reference data provides additional information on the DEM error structure, e.g. its spatial autocorrelation. If the DEM

does not contain systematic bias or alternatively a bias-corrected RMSE is provided, the unconditional stochastic simulation yields reasonable results. Otherwise, the assumptions underlying the unconditional stochastic simulation lead to an overestimation of the DEM error variability, which leads to an increased potential impact of DEM uncertainty on the potential hazard area and extreme values of dynamic flow properties. In particular, our study shows that if no higher accurate reference data is available or if computational costs of a conditional stochastic simulation would be too large,

the results of a RMSE-based unconditional stochastic simulation can still be used to provide an upper bound of the potential hazard area as well as extreme values of flow dynamic properties for a hazard assessment to take topographic uncertainties into account.





- Results of an unconditional stochastic simulation are in general sensitive to the RMSE value as well as sensitive to the fact whether or not the DEM error's spatial autocorrelation is considered. If the latter is taken into account, results are
less sensitive to actual value of the DEM error's maximum autocorrelation length. This indicates that determining a representative RMSE may be more important than finding a correct maximum autocorrelation length, while the DEM error's spatial autocorrelation should not be ignored for simulation-based landslide hazard assessment.

*Data availability.* The 5 m resolution HK-DTM used in this study is available as a free download on the website of the Survey and Mapping Office of Hong Kong under the following link: https://www.landsd.gov.hk/mapping/en/download/psi/opendata.htm. The 2 m resolution DEM
had been provided by the organizers of the Second JTC1 Workshop on Triggering and Propagation of Rapid Flow-like Landslides held in Hong Kong in 2018 to conduct a benchmark exercise. Readers can approach the Civil Engineering and Development Department of Hong Kong to obtain this DEM data for research purposes.

*Author contributions.* JK and HZ conceived the idea and designed the case study. HZ performed the implementation, simulations, data analyses with contribution from JK. HZ wrote the manuscript. JK reviewed and revised the manuscript.

*Competing interests.* The authors declare that they have no conflict of interest.

*Acknowledgements.* We gratefully acknowledge the support of Hu Zhao through the China Scholarship Council (grant number: 201706260262) and the RWTH Exploratory Research Space to further support this study. We thank the Civil Engineering and Development Department of Hong Kong for provision of the 2 m resolution DEM. We also thank DATA.GOV.HK for the public 5 m resolution DTM.



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
