# Peer review of "Topographic uncertainty quantification for flow-like landslide models via stochastic simulations"

_Natural Hazards and Earth System Sciences, 2019_

## Referee Comment (RC1) · Anonymous Referee #1 · 12 Feb 2020

The authors attempt to estimate the DEM uncertainty through stochastic simulation, and access the impacts of topographic uncertainty on simulation-based landslide run-out analyses. The subject is likely to be able to attract a broad range of the engineers and geoscientists. Overall, the article is well organized and scientifically sounds, in general. However, the 5m and 2m DTM datasets used in this study may not be the best choice. More detailed, high-accurate DTMs are recommend, such as free access Airborne LiDAR, even though the main findings by the authors may not been affected. Some minor issue listed as follows: 1. The influences/effects of elevation, slope, aspect. . ., should be clarified. 2. Table 1 is not easily to read or understand, if the manuscript is not well followed. 3. Line 218, figure 4 appeared before all the other

figures in manuscript. 4. In fig 9, the figure caption is not proper. The main findings should be noted in manuscript.

---

## Referee Comment (RC2) · Anonymous Referee #2 · 15 Feb 2020

ook laudable efforts to quantify reliability of final outputs of simulations (e.g. hazard maps). The chief subject here is topographic ones lying within pre-existing 5m HK-DTM and 2m DEM created post-event. The time difference is referred clearly (5. Line 301) and does not seem to affect the results in significant ways because of the introduction and through use of unrepresentative RMSE and subjective d. The non-affectedness itself is crucial in that every hazard map is drawn before mishaps.

[Figure]

(Specific comments) The argument depends hugely upon results obtained and shared in the second JTC1 workshop (5. Line 394), which contributes to reduce three uncertain factors other than DEM to a negligible level. Zone area and fracture height can be re-adjusted, given the very results of the authors, however. The necessity (or the negation thereof) of feed-back and of iteration in the future should be commented either in 5 or in 7.

Another minor but non-negligible issue is conditions of the channel base treated in the case study (5). The presence or absence of sizable standing trees with roots is to be mentioned 5 Line 289, given the fracture height of 1.2m.

Note:Typing errors are not discovered.

---

## Author Comment (AC2) · 12 Mar 2020

**Please note that all line numbers in the responses relate to the original manuscript.**

**Comment**: The authors took laudable efforts to quantify reliability of final outputs of simulations (e.g. hazard maps). The chief subject here is topographic ones lying within pre-existing 5 m HK-DTM and 2 m DEM created post-event. The time difference is referred clearly (5. Line 301) and does not seem to affect the results in significant ways because of the introduction and through use of unrepresentative RMSE and subjective d. The non-affectedness itself is crucial in that every hazard map is drawn before mishaps.

**Response:** We thank the reviewer for the overall positive feedback on our work. Regarding the reviewers first comment on the 'time difference', it is our impression that there had been a misunderstanding, possibly due to our narrative.

In our study, both the 5 m HK-DTM and 2 m DEM were created after the 2008 landslide event. The 2 m DEM was produced based on field mapping after the 2008 Yu Tung Road landslide event (line 295-296). The 5 m HK-DTM was generated later from a series of digital orthophotos, which were derived from aerial photographs taken in 2014 and 2015 (line 291-292). Between 2008 and 2014, hence during the time interval of data acquisition of the two DEMs, some infrastructures (debris-resisting barriers and a road, line 302) had been constructed. They are represented in the 5 m HK-DTM but not in the 2 m DEM, which leads to large inconsistency between the two DEMs in that particular area (line 303-304). Since we assumed the 2 m DEM to be more accurate than the 5 m HK-DTM and used the 2 m DEM to evaluate the error of the 5 m HK-DTM, we excluded data in this inconsistent area from higher accurate reference data. Otherwise, the error estimate of the 5 m HK-DTM may be unrealistically large (line 304-305).

We totally agree with the reviewer in that every hazard map should be drawn before mishaps, and believe that our study conception does not contrast this point. Indeed, one main conclusion of our study is that topographic uncertainty is important for simulation-based landslide hazard assessment. If a high accuracy of DEM source is not guaranteed, stochastic simulation should be conducted to provide such hazard map before mishaps so as to assess the potential hazard, rather than simply trusting results of a deterministic simulation (line 626-628). When there is available higher accuracy reference data, conditional stochastic simulation is preferred to generate such hazard map before mishaps. Otherwise, unconditional stochastic simulation can still be conducted to generate such hazard map for a hazard assessment to take topographic uncertainties into account.

To better convey our idea, we modify the sentence "It should be noted that the 2m-DEM and 5 m resolution HK-DTM were produced in different time periods. After the 2008 Yu Tung Road landslide, ..." (line 301-302) which we believe causes the misunderstanding as follows.

"It should be noted that due to different time of DEM data acquisition, there are infrastructural factors present in the 5 m resolution HK-DTM but not in the 2m-DEM. After the time of data acquisition of the 2m-DEM, ..."

**Comment:** The argument depends hugely upon results obtained and shared in the second JTC1 workshop (5. Line 394), which contributes to reduce three uncertain factors other than DEM to a negligible level. Zone area and fracture height can be re-adjusted, given the very results of the authors, however. The necessity (or the negation thereof) of feed-back and of iteration in the future should be commented either in 5 or in 7.

**Response:** We very much agree with the reviewer that not only DEM is subject to uncertainty, but release zone area, fracture height, as well as friction parameters are all potentially subject to uncertainty. All the uncertainties should be systematically quantified and the interaction between different factors should be studied eventually. In this study we decided to focus on the DEM uncertainty since this specific factor is mostly overlooked in landslide modeling and is much more complicated comparing to other uncertain factors that can be usually modeled as a probability distribution (e.g. fracture height, friction parameters, etc.). We are currently studying the relative importance of different uncertainty factors and their interaction by variance-based global sensitivity analysis. One challenge is that increasing dimension of uncertainty factors requires much larger number (tens or hundreds of thousands) of simulation runs for stochastic simulation and computational resource consuming may become prohibitively expensive. One promising solution to this challenge is to employ emulator techniques (e.g. Gaussian process emulator).

According to the reviewer's comment, the following paragraph is added after line 399 in section 5.3.1.

"It should be noted that release zone area, fracture height, as well as friction parameters may also be subject to uncertainty in landslide modeling practice. In this study we keep them fix and focus only on the DEM uncertainty which is mostly overlooked in landslide runout modeling. Future work should therefore continue to focus on systematically quantifying all the uncertainty factors and evaluating their relative importance and interaction. Researchers carrying out this work should notice that increasing dimension of uncertainty factors instantly requires much larger number of simulation runs for stochastic simulation and computational resource consuming may become prohibitively expensive. One promising solution to this challenge is to employ emulator techniques."

**Comment:** Another minor but non-negligible issue is conditions of the channel base treated in the case study (5). The presence or absence of sizable standing trees with roots is to be mentioned 5 Line 289, given the fracture height of 1.2m.

**Response:** We thank the reviewer for noticing this. The 2 m DEM is produced based on field mapping after the 2008 Yu Tung Road landslide event and reflects the bare earth at the channel base (as shown in figure 2). The 5 m resolution HK-DTM is derived from aerial photographs taken in 2014 and 2015 and includes vegetation at the channel base. Since we assume the 2m-DEM to be more accurate than the 5 m resolution HK-DTM and use the 2m-DEM to assess the error of the 5 m resolution HK-DTM, the vegetation present in the channel base in the 5 m resolution HK-DTM is not modeled

independently but treated as part of the DEM error. The HK-DTM realizations in the stochastic simulation are generated by adding the error realizations onto the 5 m resolution HK-DTM. It indicates that the channel base in the HK-DTM realizations should resemble the channel base of the 2m-DEM, which reflects the bare earth. The influence of vegetation/land cover is a very interesting topic, but is out of the scope of this study. Work in this field should be carried out in the future.

According to the reviewer's comment, the following sentences are added after line 300.

"At the channel base, the 2m-DEM reflects the bare earth and the 5 m resolution HK-DTM includes vegetation. Since we assume the 2m-DEM to be more accurate than the 5 m resolution HK-DTM and use the 2m-DEM to assess the error of the 5 m resolution HK-DTM, the vegetation present in the channel base in the 5 m HK-DTM is not modeled independently but treated as part of the DEM error."

---

## Referee Comment (RC3) · Anonymous Referee #2 · 14 Mar 2020

Reply to "Authors' reply to reviewer #2 " on "Topographic uncertainty quantification for flow-like landslide models via stochastic simulations" by Hu Zhao and Julia Kowalski Anonymous Referee #2

Overall, my initial comments, a constructive suggestion, are taken properly. The first comment was about a time difference of data acquisition, a part of which was related to the ambiguity of the timing. I mistook the 5 m HK-DTM to have been taken prior to the mishap. The authors clarified that both the 5 m HK-DTM and 2 m DEM were created after the 2008 landslide event. The point is well described in the modified sentences as caused by infrastructural factors, reflecting my first comment.

[Figure]

The second comment was about a feedback, or the mentioning thereof, of the authors' results to the 2nd JTC1 workshop participants for future revision etc. The authors chose to retell the importance of DEM uncertainty and computational challenges instead. That may be one way to pass on results to ex-participants, who will supposedly read this article. We cannot confront all uncertainties at one time methodically and for research resource binding, of course. The authors dealt the limitation issue wisely in the addendum.

The third comment was rather minor, but seems conductive for readers to draw attention to the data source acquired, field mapping as opposed to aerial photos in this case. It is really important that the inclusion of vegetation in 5 m HK-DTM to be taken note. Increase use of Lidar, replacing field survey, would provide both layers in more detail. Effects of vegetation coverage on gauge beds would be a subject in not-so-distant years.

I am grateful for being a part of review process for fruitful discussion. Thank you.

---

## Author Comment (AC3) · 21 Mar 2020

Dear reviewer #2,

We greatly appreciate your positive feedback on our response to your initial comments. Thank you very much for your time and valuable comments.

Kind regards,

Hu Zhao, on behalf on the co-authors

―――――――――――――――

2019-358, 2020.

---

## Author Response (AR1)

**Note:** This file includes: (1) response to the Editor; (2) point-by-point response to the reviewers' comments; and (3) a marked-up manuscript version showing the changes. All below-mentioned line numbers refer to the original manuscript which are consistent with the line numbers mentioned in the reviewers' comments.

**Response to the Editor**

**Editor decision:** Your manuscript has been revised and can be accepted with minor revisions for publication. Please submit a new version of the article following the referee's suggestion.

**Response:**

Dear Editor,

we greatly appreciate your time for the review process of our manuscript. Besides adjustments requested by the reviewers, we also corrected the following typos after carefully checking our manuscript.

- Change "parametrized" to "parameterized" (in line 139, line 322, and caption of figure 4).

- Change "spaial" to "spatial" (in caption of figure 4).

In addition, during the final check we came to realize that one of our responses to reviewer #2 needed further clarification. Essentially, we believe it is a minor issue, which does neither affect the results and conclusions of our study nor the essence of our response letter. In order to be transparent about any later changes, we highlighted any deviation from our original response letter. We sincerely apologize for this inconvenience.

Kind regards,

Hu Zhao, on behalf of the co-authors

**Point-by-point response to the comments of reviewer #1**

Dear reviewer #1,
We greatly appreciate your professional comments on our manuscript, which helped to improve the quality and readability of our manuscript. Our point-by-point response to each of your comments are as follows.

**Comment:** The authors attempt to estimate the DEM uncertainty through stochastic simulation and access the impacts of topographic uncertainty on simulation-based landslide run-out analyses. The subject is likely to be able to attract a broad range of the engineers and geoscientists. Overall, the article is well organized and scientifically sounds, in general. However, the $5\,m$ and $2\,m$ DTM datasets used in this study may not be the best choice. More detailed, high-accurate DTMs are recommend, such as free access Airborne LiDAR, even though the main findings by the authors may not been affected.

**Response:** Many thanks for the overall positive feedback. We totally agree with the reviewer that other high-accurate DTMs like free access Airborne LiDAR should be used eventually. High-accurate airborne LiDAR datasets are at present mostly available in developed countries, like Finland, US, Spain, etc. As stated in the introduction, "despite the broad variety of existing DEM sources, however, we are still facing (and will face in the near future) a very limited availability of high-accuracy DEMs for some regions that are particularly prone to landslide hazards, e.g. in Asia" (line 52-54). In this paper it was our goal to assess the impact of DEM uncertainty through stochastic simulation using a benchmark case that is well-known to the community. We hence decided for the 2008 Yu Tung Road landslide. To our knowledge, there was no free access Airborne LiDAR covering the area of interest when we conducted the study. Considering that the main findings may not been affected, as the reviewer points out, we still use the $5\,m$ and $2\,m$ DTM datasets for our current study that demonstrates the principal feasibility of our approach. Meanwhile, we are in active discussion with collaborators to apply our developed workflow to cases in which high accurate LiDAR datasets are available.

**Comment:** Minor issue 1. The influences/effects of elevation, slope, aspect..., should be clarified.

**Response:** We thank the reviewer for this comment. In the manuscript, terrain characteristics (e.g. elevation, slope, aspect, and ruggedness/roughness) were used for two purposes:

- to determine the required number of DEM realizations for the stochastic simulation (in section 5.2.3), and

- to analyze how DEM uncertainty affects landslide run-out modeling results (in section 6).

For the latter, we did not find obvious trends or relationships between landslide run-out modeling results and terrain characteristics at a specific location (on a cell level). One obvious reason is that a simulation result at one location is affected not only by terrain characteristics at the specific location, but by the complete upstream and surrounding terrain. We rather included a discussion of the effects of terrain by identifying several compound terrain characteristics and their impact on how DEM uncertainty may affect landslide run-out modeling results. The compound terrain characteristics include: a) banks of the channel, especially the north bank near area 1 and south bank near area 2 (line 473, line 479-480, figure 8); b) relatively high elevation area at the end part of the channel that holds back flow material (line 473-474); c) topographic roughness (line 474); d) relatively flat area 3 (line 482, figure 8).
Due to DEM uncertainty, above compound terrain characteristics represented in DEM realizations vary with respect to the original DEM (line 471-472), which further affects landslide run-out modeling results. Specifically,

a) tend to be dampened out from DEM realizations (line 472-474). Deteriorated channel bank representation makes flow material spread out along channel cross section direction, travel shorter distance (line 485-486), and leads to smaller ensemble-based mean of flow dynamic properties at channel bottom locations (line 543-544).

b) also tends to be dampened out from DEM realizations (line 472-474), which makes flow material travel further (line 486-487).

c) tends to increase (line 474), which leads to higher simulated momentum losses, shorter travel distance (line 487-488), and smaller flow dynamic properties on average (line 550).

d) area 3 is relatively flat (namely, slopes in the area are relatively small). Thus, it is sensitive to DEM uncertainty (line 482-483). Furthermore, it locates near the deposition, around which the impact of upstream DEM uncertainty seems to accumulate (line 496-497). Both explain the impact of DEM uncertainty in that area.

As a summary, we discussed the effects of terrain characteristics on simulation results in terms of compound terrain characteristics (e.g. banks of the channel, relatively high elevation area at the end part of the channel, topographic roughness, and relatively flat/small slope area 3) rather than the effects of terrain characteristics at the cell level (e.g. elevation, slope, ruggedness/roughness at a specific location). As regards to aspect, we did not find a clear relationship between this terrain characteristic and simulation results. Therefore, we did not discuss it in section 6.

We agree that our text can be improved to convey our ideas and improve the readability. We therefore modify our manuscript as follows.

- To avoid confusion, we keep the terminology "roughness" consistent throughout the manuscript. Namely, we change all the word "ruggedness" to "roughness" (in line 248, line 272, line 355, line 361, line 364, and legend of figure 5).

- We add the following paragraph before we start the discussion (after line 470) to explain why we discuss the effects of terrain characteristics on simulation results in terms of above-mentioned compound terrain characteristics.

"As stated in section 4, analyzing terrain characteristics of the original DEM and DEM realizations may help us to interpret simulation results. By a preliminary analysis, we did not find obvious relationships between landslide run-out simulation results and terrain characteristics at a specific location (on the cell level). One obvious reason is that a simulation result at one location is affected not only by terrain characteristics at the specific location, but by the complete upstream and surrounding terrain. Instead of discussing the effects of terrain characteristics at the cell level, we therefore focus on several compound terrain characteristics that help us to understand how DEM uncertainty may affect simulation results. The compound terrain characteristics include: banks of the channel, especially the north bank near area 1 and south bank near area 2; relatively high elevation area at the end part of the channel that holds back flow material as shown in Fig. 7 (b); topographic roughness; relatively flat area 3 (namely area with relatively small slope)."

The following sentence and content are accordingly removed to keep the manuscript coherent and concise: "This can be explained as follows." (line 469-470) and "as shown in Fig. 7 (b)" (line 474).

**Comment:** Minor issue 2. Table 1 is not easily to read or understand, if the manuscript is not well followed.

**Response:** we thank the reviewer for pointing this out. To improve the readability, we modify Table 1 as follows.

**Table 1.** Scenarios for stochastic simulation

| | method to generate DEM realizations | input to generate DEM realizations |
|---|---|---|
| scenario A) | USS | RMSE=3.3; d=180 |
| scenario B) | CSS | semivariogram; $\epsilon^*_{KK}\{K = 180\}$ |
| scenarios for *unrepresentative RMSE* | USS | RMSE=0.5, 1.5, 2.5; d=180 |
| scenarios for *subject d* | USS | RMSE=3.3; d=0, 90, 270 |

Note: in scenario A) and B), the inputs to generate DEM realizations are obtained from higher accurate reference data at the 180 reference locations.

**Comment:** Minor issue 3. Line 218, figure 4 appeared before all the other figures in manuscript.

**Response**: We thank the reviewer for noticing this. We originally referred to figure 4 in line 218 to give the readers who are not familiar with semivariogram models a direct impression. To avoid confusion, we replace the notation "(see Fig. 4)" with "(see section 5.2.1)" in line 218.

**Comment:** Minor issue 4. In fig 9, the figure caption is not proper. The main findings should be noted in manuscript.

**Response:** We thank the reviewer for this comment. To avoid redundancy, we remove the main findings in the caption of figure 9 according to the reviewer's comment. We accordingly remove the main findings in the caption of figure 8 as well. Both have already been noted in the manuscript (line 556-560 for figure 9 and line 507-510 for figure 8).

<h1 style="text-align:center">Revised Point-by-point response to the comments of reviewer #2</h1>

Dear reviewer #2,

We thank you for taking the time to read and comment on our manuscript. We believe the comments help to improve the quality of our manuscript and lead to a much clearer revised version of the paper. We also greatly appreciate your positive feedback on our reply to your initial comments.

While assembling this final version of the manuscript, we carefully checked all associated files and information again, and came to the conclusion that we have to complete and slightly correct our own response to your review:

In our response letter we wrote that the 2m-DEM was produced based on field mapping after the 2008 Yu Tung Road landslide event (line 295-296). Reconsideration of the "note to participants" of the Second JTC1 Workshop (http://www.hkges.org/ JTC1_2nd/be.html) yielded that the 2m-DEM is in fact the result of merging a pre-event DEM with data from a detailed field mapping in the release area after the 2008 landslide event (AECOM Asia Company Limited, 2012). In order to avoid any misunderstanding, we hence modified and extended the manuscript to better reflect the *merged character* of the DEM.

While we believe that the necessary adjustments neither affect the results and conclusions of this study, nor the essence of our reply to your review, we did have to change some of our concrete responses and highlighted them in  and blue.

We sincerely apologize for not expressing the information clearly in the first place, and any inconvenience this might have caused.

**General comment:** The authors took laudable efforts to quantify reliability of final outputs of simulations (e.g. hazard maps). The chief subject here is topographic ones lying within pre-existing $5\,m$ HK-DTM and $2\,m$ DEM created post-event. The time difference is referred clearly (5. Line 301) and does not seem to affect the results in significant ways because of the introduction and through use of unrepresentative RMSE and subjective d. The non-affectedness itself is crucial in that every hazard map is drawn before mishaps.

**Response:** We thank the reviewer for the overall positive feedback on our work. Regarding the reviewers first comment on the 'time difference',  there had been a misunderstanding  due to our narrative.

In our study, the $2\,m$ DEM was produced based on field mapping after the 2008 Yu Tung Road landslide event  and a pre-event DEM. Specifically, the 2m-DEM reflects the rupture surface in the release zone area and reflects the pre-event slope surface in other areas. The $5\,m$ HK-DTM was generated later from a series of digital orthophotos, which were derived from aerial photographs taken in 2014 and 2015 (line 291-292). Between 2008 and 2014, hence during the time interval of data acquisition of the two DEMs, some infrastructures (debris-resisting barriers and a road, line 302) had been constructed. They are represented in the $5\,m$ HK-DTM but not in the $2\,m$ DEM, which leads to large inconsistency between the two DEMs in that particular area (line 303-304). Since we assumed the $2\,m$ DEM to be more accurate than the $5\,m$ HK-DTM and used the $2\,m$ DEM to evaluate the error of the $5\,m$ HK-DTM, we excluded data in this inconsistent area from higher accurate reference data. Otherwise, the error estimate of the $5\,m$ HK-DTM may be unrealistically large (line 304-305).

We totally agree with the reviewer in that every hazard map should be drawn before mishaps, and believe that our study conception does not contrast this point. Indeed, one main conclusion of our study is that topographic uncertainty is important for simulation-based landslide hazard assessment. If a high accuracy of DEM source is not guaranteed, stochastic simulation should be conducted to provide such hazard map before mishaps so as to assess the potential hazard, rather than simply trusting results of a deterministic simulation (line 626-628). When there is available higher accuracy reference data, conditional stochastic simulation is preferred to generate such hazard map before mishaps. Otherwise, unconditional stochastic simulation can still be conducted to generate such hazard map for a hazard assessment to take topographic uncertainties into account.

To better convey our idea, we  make adjustments as follows.

- modify "It should be noted that the 2m-DEM and $5\,m$ resolution HK-DTM were produced in different time periods. After the 2008 Yu Tung Road landslide, …" (line 301-302) to

v

"It should be noted that due to different time of DEM data acquisition, there are infrastructural factors present in the 5 $m$ resolution HK-DTM but not in the 2m-DEM. After the time of data acquisition of the 2m-DEM, ..."

- modify "It is produced based on field mapping after the 2008 Yu Tung Road landslide event" (line 295-296) to

"It is produced based on the field mapping after the 2008 Yu Tung Road landslide event and a pre-event DEM. According to the "note to participants" of the Second JTC1 Workshop (which can be found under the link: http://www.hkges.org/JTC1_2nd/be.html), the 2m-DEM represents the rupture surface in the release zone area and the pre-event slope surface in other areas. The rupture surface is obtained based on the field mapping (AECOM Asia Company Limited, 2012)."

**Specific comment 1:** The argument depends hugely upon results obtained and shared in the second JTC1 workshop (5. Line 394), which contributes to reduce three uncertain factors other than DEM to a negligible level. Zone area and fracture height can be re-adjusted, given the very results of the authors, however. The necessity (or the negation thereof) of feed-back and of iteration in the future should be commented either in 5 or in 7.

**Response:** We very much agree with the reviewer that not only DEM is subject to uncertainty, but release zone area, fracture height, as well as friction parameters are all potentially subject to uncertainty. All the uncertainties should be systematically quantified and the interaction between different factors should be studied eventually. In this study we decided to focus on the DEM uncertainty since this specific factor is mostly overlooked in landslide modeling and is much more complicated comparing to other uncertain factors that can be usually modeled as a probability distribution (e.g. fracture height, friction parameters, etc.). We are currently studying the relative importance of different uncertainty factors and their interaction by variance-based global sensitivity analysis. One challenge is that increasing dimension of uncertainty factors requires much larger number (tens or hundreds of thousands) of simulation runs for stochastic simulation and computational resource consuming may become prohibitively expensive. One promising solution to this challenge is to employ emulator techniques (e.g. Gaussian process emulator).
According to the reviewer's comment, the following paragraph is added after line 399 in section 5.3.1.

"It should be noted that release zone area, fracture height, as well as friction parameters may also be subject to uncertainty in landslide modeling practice. In this study we keep them fix and focus only on the DEM uncertainty which is mostly overlooked in landslide run-out modeling. Future work should therefore continue to focus on systematically quantifying all the uncertainty factors and evaluating their relative importance and interaction. Researchers carrying out this work should notice that increasing dimension of uncertainty factors instantly requires much larger number of simulation runs for stochastic simulation and computational resource consuming may become prohibitively expensive. One promising solution to this challenge is to employ emulator techniques."

**Specific comment 2:** Another minor but non-negligible issue is conditions of the channel base treated in the case study (5). The presence or absence of sizable standing trees with roots is to be mentioned 5 Line 289, given the fracture height of 1.2m.

**Response:** We thank the reviewer for noticing this.  The 2m-DEM reflects the rupture surface in the release zone area and reflects the pre-event slope surface in other areas. We do not have information about the vegetation on its channel base. The 5 $m$ resolution HK-DTM is derived from aerial photographs taken in 2014 and 2015 and includes vegetation at the channel base. ~~Since we assume the 2m-DEM to be more accurate than the 5 $m$ resolution HK-DTM and use the 2m-DEM to assess the error of the 5 $m$ resolution HK-DTM, the vegetation present in the channel base in the 5 $m$ resolution HK-DTM is not modeled independently but treated as part of the DEM error. The HK-DTM realizations in the stochastic simulation are generated by adding the error realizations onto the 5 $m$ resolution HK-DTM. It indicates that the channel base in the HK-DTM realizations should resemble the channel base of the 2m-DEM, which reflects the bare earth.~~

In this study, any vegetation present in the channel base in the $5\,m$ resolution HK-DTM is not explicitly accounted for in the sense of a modeled DTM correction. It is rather subsumed as part of the DEM error. The influence of vegetation/land cover is a very interesting topic, but is out of the scope of this study. Work in this field should be carried out in the future.

According to the reviewer's comment, the following sentences are added after line 300.

[revised manuscript text omitted]